# Universality of Many-body Projected Ensemble for Learning Quantum Data Distribution

## Abstract

Generating quantum data by learning the underlying quantum distribution poses challenges in both theoretical and practical scenarios, yet it is a critical task for understanding quantum systems. A fundamental question in quantum machine learning (QML) is the universality of approximation: whether a parameterized QML model can approximate any quantum distribution. We address this question by proving a universality theorem for the Many-body Projected Ensemble (MPE) framework, a method for quantum state design that uses a single many-body wave function to prepare random states. This demonstrates that MPE can approximate any distribution of pure states within a 1-Wasserstein distance error. This theorem provides a rigorous guarantee of universal expressivity, addressing key theoretical gaps in QML. For practicality, we propose an Incremental MPE variant with layer-wise training to improve the trainability. Numerical experiments on clustered quantum states and quantum chemistry datasets validate MPE's efficacy in learning complex quantum data distributions.

## 1 Introduction

Recent advancements highlight the pivotal role of quantum machine learning (QML) (Dunjko et al., 2016; Biamonte et al., 2017) in processing quantum data derived from quantum systems (Editorial, 2023). A fundamental task in QML is generating quantum data by learning the underlying distribution, essential for understanding quantum systems, synthesizing new samples, and advancing applications in quantum chemistry and materials science. However, extending classical generative approaches to quantum data presents significant challenges, as quantum distributions exhibit superposition, entanglement, and non-locality that classical models struggle to replicate efficiently.

Quantum generative models such as quantum generative adversarial networks (Lloyd & Weedbrook, 2018; Zoufal et al., 2019) and quantum variational autoencoders (Khoshaman et al., 2018; Wu et al., 2024) can be used to prepare a fixed single quantum state (Niu et al., 2022; Kim et al., 2024; Tran et al., 2024), but are inefficient for generating ensembles of quantum states (Beer & Müller, 2021) due to the need for training deep parameterized quantum circuits (PQCs). The quantum denoising diffusion probabilistic model (Zhang et al., 2024) offers a promising approach that employs intermediate training steps to smoothly interpolate between the target distribution and noise, thereby enabling efficient training. However, the diffusion process requires high-fidelity scrambling random unitary circuits, demanding implementation challenges of precise spatio-temporal control.

Learning quantum data distributions faces significant hurdles in the noisy intermediate-scale quantum (NISQ) era, including noise-induced errors, limited qubit connectivity, and optimization difficulties such as barren plateaus (McClean et al., 2018), where gradients vanish exponentially with system size. Moreover, achieving universality, which entails the model's ability to approximate any quantum distribution with arbitrary precision, remains a significant theoretical and practical challenge. These limitations underscore the need for innovative frameworks that combine theoretical guarantees of universality with scalable, noise-resilient training strategies.

In this study, we prove a universality theorem for learning quantum data distributions. Our proof relies on the Many-body Projected Ensemble (MPE) framework, a recent approach in quantum state design that uses a single many-body wave function to prepare random states (Choi et al., 2023; Cotler et al., 2023). We demonstrate that MPE can approximate any $n$-qubit pure state distribution within a

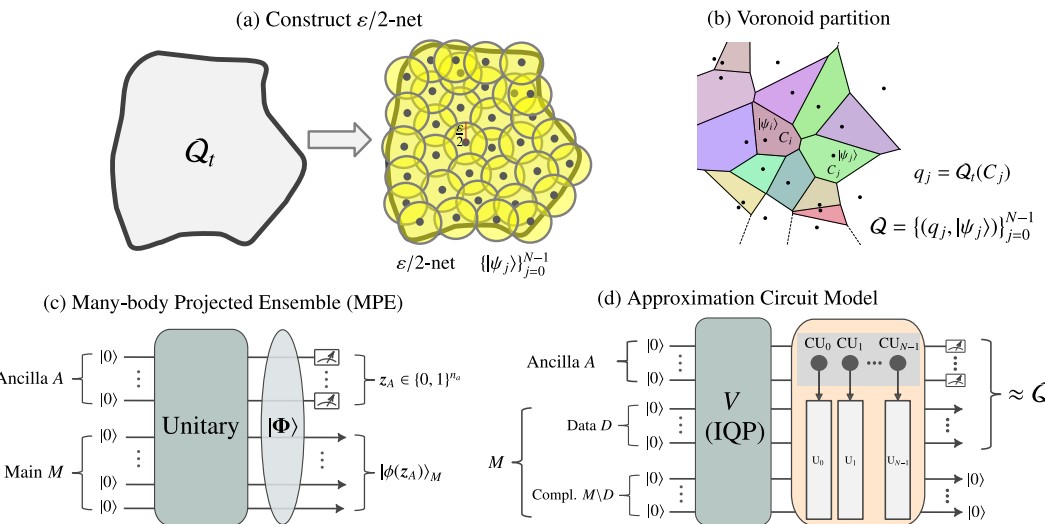

Figure 1: A scheme to construct a parameterized quantum distribution $\mathcal{Q}_{\boldsymbol{\theta}}$ approximating a target $\mathcal{Q}_t$ within error $\varepsilon$. (a) Form an $\varepsilon/2$-net ensemble to approximate $\mathcal{Q}_t$ within $\varepsilon/2$, with probabilities from (b) Voronoi partitioning. (c) Perform partial measurements of a single many-body wave function $|\boldsymbol{\Phi}\rangle$ on ancilla system $A$, yielding a projected ensemble in main system $M$. (d) Implement the approximation using an IQP circuit and controlled unitary circuits for the projected ensembles.

specified 1-Wasserstein distance error bound, leveraging covering discretization for the target quantum distribution and ancilla-assisted measurements to represent the discrete ensembles. This result is a cornerstone for the QML community, providing a rigorous theoretical guarantee of universal expressivity, enabling the modeling of complex quantum distributions in applications like quantum chemistry. While practical QML methods face issues such as classical simulability, barren plateaus, and high resource requirements in training (Appendix A.3), we view the universality theorem as complementary, ensuring that models can, in principle, capture any distribution before optimizing for hardware. To facilitate practical implementation, we introduce an Incremental MPE variant that employs layer-wise training to improve trainability and reduce resource demands, thereby enhancing compatibility with NISQ devices. Our numerical experiments on clustered quantum states and computational chemistry datasets validate the efficacy of the framework.

## 2 LEARNING QUANTUM DATA DISTRIBUTION

Generative models are powerful tools for generating samples from a target distribution and estimating the likelihood of given data points. We address the problem of learning an unknown quantum data distribution $\mathcal{Q}_t$ over $n$-qubit pure states, given a training dataset $\mathcal{S} = \{|\psi_0\rangle, \ldots, |\psi_{N-1}\rangle\}$ of $N$ independent states sampled from $\mathcal{Q}_t$. The generative model is defined by a parameterized probability distribution $\mathcal{Q}_{\boldsymbol{\theta}}$ implemented via PQCs, where $\boldsymbol{\theta}$ represents the trainable parameters (e.g., gate angles). The training objective is to optimize $\boldsymbol{\theta}$ such that $\mathcal{Q}_{\boldsymbol{\theta}}$ closely approximates $\mathcal{Q}_t$, as measured by a distance metric $\mathcal{D}(\mathcal{Q}_{\boldsymbol{\theta}}, \mathcal{Q}_t)$. Since directly computing $\mathcal{D}(\mathcal{Q}_{\boldsymbol{\theta}}, \mathcal{Q}_t)$ is often infeasible, we sample a dataset $\tilde{\mathcal{S}} = \{|\tilde{\psi}_j\rangle\}_j$ from $\mathcal{Q}_{\boldsymbol{\theta}}$ and minimize the empirical distance $\mathcal{D}(\mathcal{S}, \tilde{\mathcal{S}})$. In the inference phase, the optimized parameters $\boldsymbol{\theta}_{\text{opt}}$ are fixed, and new quantum states $|\psi\rangle \sim \mathcal{Q}_{\boldsymbol{\theta}_{\text{opt}}}$ are generated for use in quantum simulation and data analysis.

## 3 PRELIMINARIES AND PROBLEM FORMULATION

### 3.1 DISTANCE BETWEEN DISTRIBUTIONS

For a density operator $\rho$ acting on a Hilbert space (see Appendix A.1), we define the trace norm as $\|\rho\|_1 = \text{Tr}\sqrt{\rho^\dagger \rho}$, where $\text{Tr}$ denotes the trace operation and $\rho^\dagger$ is the Hermitian conjugate of $\rho$. The trace distance between two density operators $\rho$ and $\sigma$ is $d(\rho, \sigma) = \frac{1}{2}\|\rho - \sigma\|_1$. This distance metric

captures the distinguishability of two quantum states and serves as a fundamental measure in quantum information theory. To compare ensembles of quantum states, we employ the 1-Wasserstein distance, which extends the trace distance to probability distributions over density operators.

**Definition 3.1** (1-Wasserstein Distance). *Let $P$ and $Q$ be two probability measures (or ensembles) over the space of density operators. The 1-Wasserstein distance between $P$ and $Q$ is defined as the minimal expected trace distance between pairs of states sampled from a coupling of $P$ and $Q$:*

$$W_1(P, Q) = \inf_{\pi \in \Pi(P,Q)} \mathbb{E}_{(\rho,\sigma) \sim \pi} \left[ \frac{1}{2} \|\rho - \sigma\|_1 \right], \tag{1}$$

*where $\Pi(P, Q)$ denotes the set of couplings (joint probability measures) with marginals $P$ and $Q$.*

### 3.2 Many-body Projected Ensemble (MPE)

An ensemble of states can be generated from a single wave function by performing local measurements over a part of the total system. We consider many-body system partitioned into a subsystem $M$ (with $n_m$ qubits) and its complement $A$ (with $n_a$ qubits). For the unification in this manuscript, we consider $A$ as the ancillary system [Fig. 1(c)]. Given a generator state $|\Phi\rangle$, which is a pure many-body wave function on the total system $A + M$, we perform local measurements on $A$, typically in the computational basis. This yields different pure states $|\phi(z_A)\rangle_M$ on $M$, each corresponding to a distinct measurement outcome $z_A$ on $A$, which are bitstrings of the form, for example, $z_A = 001 \dots 010$. The collection of these states, together with probabilities $p(z_A)$, forms the many-body projected ensemble (MPE) on $M$: $\{(p(z_A), |\phi(z_A)\rangle_M)\}_{z_A}$. The projected ensemble provides a full description of the total system state as $|\Phi\rangle_{A+M} = \sum_{z_A \in \{0,1\}^{n_a}} \sqrt{p(z_A)} |z_A\rangle \otimes |\phi(z_A)\rangle_M$. MPE is used to approximate a Haar-random state ensemble (Choi et al., 2023; Cotler et al., 2023), providing insights into the study of complexity growth in quantum systems (Appendix A.2). In our study, MPE is used to prove the universality with the potential to yield an advantage in generative models, as classical methods struggle to prepare the many-body state.

### 3.3 Problem Formulation

Given a target distribution $\mathcal{Q}_t$ over pure $n$-qubit states, our objective is to propose a class of parameterized probability distribution $\mathcal{Q}_\theta$, where $\theta$ denotes the parameters of the model, such that for any error $\varepsilon > 0$, there exists $\theta$ satisfying $W_1(\mathcal{Q}_t, \mathcal{Q}_\theta) \leq \varepsilon$. We assume $\mathcal{Q}_t$ is unknown but samples from $\mathcal{Q}_t$ are available for training. This problem is central to generative QML as it addresses the challenge of accurately approximating complex quantum data distributions with applications in quantum simulation, quantum chemistry, and quantum information processing.

## 4 Main Result: Universality Approximation Theorem

In approximating arbitrary $n$-qubit quantum data distributions, we propose a systematic method that combines discretization techniques with MPE. The procedure consists of the following steps.

**Discretizing the target distribution with an $\varepsilon/2$-covering technique** [Fig. 1(a)(b)]: This involves constructing a finite set of quantum states (an $\varepsilon/2$-net) such that every state in $\mathcal{Q}_t$ is within a trace distance of at most $\varepsilon/2$ from at least one state in the $\varepsilon/2$-net. This step yields a discrete distribution that approximates $\mathcal{Q}_t$ within an error bound of $\varepsilon/2$.

**Applying the MPE** [Fig. 1(c)(d)]: This constructs $\mathcal{Q}_\theta$ by leveraging partial measurements to approximate the discrete $\varepsilon/2$-net within an error bound of $\varepsilon/2$.

The use of $\varepsilon$-covering ensures computational feasibility for continuous space, while the MPE framework leverages the structure of many-body quantum systems to achieve high-fidelity approximations. Here, we state the following universality theorem for the MPE framework:

**Theorem 4.1** (Universality Approximation of the Many-body Projected Ensemble). *For any target quantum data distribution $\mathcal{Q}_t$ over pure $n$-qubit states, there exists a parameterized quantum distribution $\mathcal{Q}_\theta$, formulated through the Many-body Projected Ensemble (MPE) framework utilizing a covering technique and ancilla-assisted measurements, such that for any error bound $\varepsilon > 0$, there exists a parameter $\theta^*$ for which the 1-Wasserstein distance satisfies $W_1(\mathcal{Q}_t, \mathcal{Q}_{\theta^*}) \leq \varepsilon$.*

*Proof.* The proof constructs $\mathcal{Q}_{\boldsymbol{\theta}}$ using the two steps above, supported by Lemmas 4.2, 4.3, and 4.5. Lemma 4.2 establishes a discrete ensemble $\mathcal{Q} = \{(q_j, |\psi_j\rangle)\}_{j=0}^{N-1}$ such that $W_1(\mathcal{Q}_t, \mathcal{Q}) \leq \varepsilon/2$. Next, the MPE framework (Lemma 4.3 or Lemma 4.4) constructs a class of projected ensemble $\mathcal{P} = \{(p_j, |\psi_j\rangle)\}_{j=0}^{N-1}$ identified as $\mathcal{Q}_{\boldsymbol{\theta}}$ such that there exist a parameter $\boldsymbol{\theta}^*$ for which $W_1(\mathcal{Q}_{\boldsymbol{\theta}^*}, \mathcal{Q}) \leq \varepsilon/2$. By the triangle inequality, we have $W_1(\mathcal{Q}_t, \mathcal{Q}_{\boldsymbol{\theta}^*}) \leq W_1(\mathcal{Q}_t, \mathcal{Q}) + W_1(\mathcal{Q}, \mathcal{Q}_{\boldsymbol{\theta}^*}) \leq \frac{\varepsilon}{2} + \frac{\varepsilon}{2} = \varepsilon$. The scaling of $N$ and the number of ancilla qubits are detailed in the lemmas below. □

### 4.1 $\varepsilon/2$-Covering for Quantum State Distribution Discretization

To enable the approximation of a quantum data distribution ensuring the usage of the MPE framework, we first introduce a key lemma that establishes the existence of a finite ensemble of pure quantum states approximating a target distribution within a 1-Wasserstein distance of $\varepsilon/2$.

**Lemma 4.2** (Finite Ensemble Approximation)**.** *For any target quantum data distribution $\mathcal{Q}_t$ over pure $n$-qubit states and any $\varepsilon > 0$, there exists a finite ensemble $\mathcal{Q} = \{(q_j, |\psi_j\rangle)\}_{j=0}^{N-1}$ such that the 1-Wasserstein distance satisfies $W_1(\mathcal{Q}_t, \mathcal{Q}) \leq \frac{\varepsilon}{2}$.*

*Proof.* Let $\delta = \varepsilon/2$. The trace distance between two pure quantum states $|\psi\rangle$ and $|\phi\rangle$ is defined as $d(|\psi\rangle, |\phi\rangle) = \frac{1}{2} \||\psi\rangle\langle\psi| - |\phi\rangle\langle\phi|\|_1$.

**Existence of a finite $\delta$-net** [Fig. 1(a)]: By compactness, there exists a finite set of pure states $\{|\psi_j\rangle\}_{j=0}^{N-1}$ (a $\delta$-net) such that for every $|\psi\rangle \sim \mathcal{Q}_t$ there is some $|\psi_j\rangle$ satisfying $d(|\psi\rangle, |\psi_j\rangle) \leq \delta$. This $\delta$-net forms the basis for discretizing the state space. Here, $N$ is the $\delta$-covering number of $(\mathcal{Q}_t, d)$, which is the cardinality $\mathcal{N}(\mathcal{Q}_t, d, \delta)$ of the smallest $\delta$-net of $\mathcal{Q}_t$. If we consider $\mathcal{Q}_t$ is the $D$-dimensional subspace of the full Hilbert space $\mathbb{C}^{2^n}$ ($D \geq 2$), standard results from high-dimensional geometry and geometry of quantum states provide an upper bound for $\mathcal{N}(\mathcal{Q}_t, d, \delta)$ as follows (see Appendix A.4 for the construction of $\delta$-net and the formal proof for the bound of covering number):

$$N = \mathcal{N}(\mathcal{Q}_t, d, \delta) \leq 5 \cdot D \ln(D) \cdot (1/\delta)^{2(D-1)}. \tag{2}$$

**Voronoi partition** [Fig. 1(b)]: We define measurable cells $\{C_j\}_{j=0}^{N-1}$ that partition the space of pure states as $C_j = \{|\psi\rangle : d(|\psi\rangle, |\psi_j\rangle) \leq d(|\psi\rangle, |\psi_i\rangle)$ for all $i\}$, where $|\psi_j\rangle$ is the center of $C_j$. By the property of the $\delta$-net, for every $|\psi\rangle \in C_j$ we have $d(|\psi\rangle, |\psi_j\rangle) \leq \delta$. These cells form a Voronoi partition assigning each state to the nearest center in the $\delta$-net. We define $q_j = \mathcal{Q}_t(C_j)$, which is the probability that $\mathcal{Q}_t$ assigns to all pure states in cell $C_j$. The finite ensemble is defined as $\mathcal{Q} = \{(q_j, |\psi_i\rangle)\}_{j=0}^{N-1}$, representing a discrete approximation of $\mathcal{Q}_t$ with each $|\psi_j\rangle$ weighted by the probability mass of its corresponding cell.

**Bounding the 1-Wasserstein Distance**: We construct an explicit coupling $\pi \in \Pi(\mathcal{Q}_t, \mathcal{Q})$. For each state $|\psi\rangle \sim \mathcal{Q}_t$, identify the unique index $j$ such that $|\psi\rangle \in C_j$ and pair the density operator $|\psi\rangle\langle\psi|$ with $|\psi_j\rangle\langle\psi_j|$. The 1-Wasserstein distance is then bounded as

$$W_1(\mathcal{Q}_t, \mathcal{Q}) \leq \mathbb{E}_{|\psi\rangle \sim \mathcal{Q}_t}[d(|\psi\rangle, |\psi_j\rangle)] \leq \delta = \frac{\varepsilon}{2}. \tag{3}$$

Equation 3 completes the proof, where the second inequality holds because for each $|\psi\rangle \in C_j$ the trace distance to the center $|\psi_j\rangle$ is less than $\delta$ by the property of the $\delta$-net and Voronoi partition. □

### 4.2 Applying the Many-body Projected Ensemble (MPE)

In our setting, the target distribution $\mathcal{Q}_t$ is unknown, but samples from $\mathcal{Q}_t$ are available for training. The $\varepsilon/2$-covering technique provides the explicit construction of a $\delta$-net $\{|\psi_j\rangle\}_{j=0}^{N-1}$ (Appendix A.4), though the probabilities $\{q_j\}_{j=0}^{N-1}$ remain unknown. The next step constructs a class of PQCs to produce a projected ensemble $\mathcal{P} = \{(p_j, |\psi_j\rangle)\}_{j=0}^{N-1}$ such that the probability distribution $p = \{p_j\}_j$ approximating the target distribution $q = \{q_j\}_j$. This step leverages a composite quantum system comprising an ancilla system $A$ with $n_a = \lceil \log_2 N \rceil$ qubits (ensuring $N \leq 2^{n_a}$), and a hidden system $M$ with $n_m$ qubits. The following lemmas establish the existence and construction of $V$ acting on $A \otimes M$ that generates $\{p_j\}_j \approx \{q_j\}_j$. We use $\boldsymbol{j} \in \{0, 1\}^{n_a}$ to denote the measurement outcome (binary string), and $j \in \{0, \ldots, 2^{n_a} - 1\}$ is the decimal equivalent of binary string $\boldsymbol{j}$.

**Lemma 4.3** (Approximate Probability Distribution). *Given a target probability distribution $q = \{q_j\}_{j=0,1,\ldots,2^{n_a}-1}$ and an error bound $\varepsilon > 0$, there exists a parameterized unitary $V$ acting on the ancilla system $A$ (with $n_a = \lceil \log_2 N \rceil$ qubits) and a hidden system $M$ (with $n_m = n_a + \lceil \log_2(1/\varepsilon) \rceil$ qubits) such that after applying $V$ and measuring $A$ in the computational basis, the resulting probability distribution $p = \{p_j\}_{j=0,1,\ldots,2^{n_a}-1}$ satisfies $\delta(p,q) \leq \frac{\varepsilon}{2}$, where the total variation distance is defined as $\delta(p,q) = \frac{1}{2}\sum_j |p_j - q_j|$. Here, $p_j = p(\boldsymbol{j})$ is the probability to obtain the measure outcome $\boldsymbol{j} \in \{0,1\}^{n_a}$ in $A$. Furthermore, an explicit construction of $V$ is provided, extending the result of Lemma 1 in* Kurkin et al. (2025).

*Proof Sketch.* The proof extends Lemma 1 in Kurkin et al. (2025), which establishes the existence of an Instantaneous Quantum Polynomial (IQP) (Shepherd & Bremner, 2009; Bremner et al., 2010) circuit $V$ satisfying $\delta(p,q) \leq \varepsilon/2$ with $n_m = n_a + \lceil \log_2(1/\varepsilon) \rceil$. We propose a specific implementation of $V$ detailed in Appendix 4.3. $\square$

As detailed in Appendix 4.3, Lemma 4.3 constructs an approximate $p \approx q$ using the IQP circuits with total $2^{n_a+n_m}$ parameters, but restricted to only two real values $0$ and $\pi$. The following lemma ( Lemma 5 in Kurkin et al. (2025)) achieves exact $p = q$ with the same total parameters, but the parameters are full complex values. The idea is to decompose $q$ into mixtures of 2-sparse distributions with $n_m = n_a + 1$ hidden qubits and complex phases. The construction is independent of error, as it is exact, but complex phases increase parameter expressivity and training costs.

**Lemma 4.4** (Exact Probability Distribution). *There exists a parameterized unitary $V$ acting on the ancilla system $A$ (with $n_a = \lceil \log_2 N \rceil$ qubits) and a hidden system $M$ (with sufficiently many qubits $n_m$) such that after applying $V$ and measuring $A$ in the computational basis, the resulting probability distribution $p = \{p_j\}_{j=0,1,\ldots,2^{n_a}-1}$ exactly matches the target distribution, i.e., $p = q$.*

After applying $V$ on the composite system $A \otimes M$ initialized in the state $|\mathbf{0}\rangle_A |\mathbf{0}\rangle_M$, we obtain a projected ensemble $\{(p(\boldsymbol{j}), |\phi(\boldsymbol{j})\rangle_M)\}_{\boldsymbol{j} \in \{0,1\}^{n_a}}$, where $|\phi(\boldsymbol{j})\rangle_M$ is the state in the hidden system $M$ corresponding with the measurement outcome $\boldsymbol{j}$. This process is summarized as follows:

$$|0\rangle_A |0\rangle_M \xrightarrow{V} \sum_{\boldsymbol{j} \in \{0,1\}^{n_a}} \sqrt{p(\boldsymbol{j})} |\boldsymbol{j}\rangle_A \otimes |\phi(\boldsymbol{j})\rangle_M. \tag{4}$$

We select $M$ such that the number $n_m$ of qubits in $M$ is larger than the number $n_d$ of data qubits. Then $M$ can be divided into the data system $D$ and the complementary system $M \backslash D$.

Next, we apply a series of multi-qubits controlled unitaries $\text{CU}_0, \text{CU}_1, \ldots$, such that if the measurement outcome in $A$ is $\boldsymbol{j}$, the unitary $U_j$ will transform the state $|\phi(\boldsymbol{j})\rangle_M$ to $|\psi_j\rangle_D \otimes |\mathbf{0}\rangle_{M \backslash D}$ [Fig. 1(d)]. This operation is described as:

$$\sum_{\boldsymbol{j} \in \{0,1\}^{n_a}} \sqrt{p(\boldsymbol{j})} |\boldsymbol{j}\rangle_A \otimes |\phi(\boldsymbol{j})\rangle_M \xrightarrow{\Pi_{j=0}^{2^{n_a}-1} \text{CU}_j} \sum_{j=0}^{2^{n_a}-1} \sqrt{p_j} |j\rangle_A \otimes |\psi_j\rangle_D \otimes |\mathbf{0}\rangle_{M \backslash D}, \tag{5}$$

where $p_j = p(\boldsymbol{j})$. The resulting ensemble obtained from the data system $D$ is $\mathcal{P} = \{(p_j, |\psi_j\rangle)\}_{j=0}^{N-1}$, designed to approximate $\mathcal{Q} = \{(q_j, |\psi_j\rangle)\}_{j=0}^{N-1}$ via the following lemma.

**Lemma 4.5** (Wasserstein Distance Bound for Projected Ensemble). *The projected ensemble $\mathcal{P} = \{(p_j, |\psi_j\rangle)\}_{j=0}^{N-1}$ constructed via the MPE framework (using Lemma 4.3 or Lemma 4.4, and Equation 5) and the discrete ensemble $\mathcal{Q} = \{(q_j, |\psi_j\rangle)\}_{j=0}^{N-1}$ from Lemma 4.2 satisfy $W_1(\mathcal{P}, \mathcal{Q}) \leq \frac{\varepsilon}{2}$.*

*Proof.* By Lemma 4.3, the projected ensemble $\mathcal{P} = \{(p_j, |\psi_j\rangle)\}_{j=0}^{N-1}$ satisfies the total variation distance bound $\delta(p,q) = \frac{1}{2}\sum_{j=0}^{N-1} |p_j - q_j| \leq \frac{\varepsilon}{2}$, where $p = \{p_j\}_{j=0}^{N-1}$ and $q = \{q_j\}_{j=0}^{N-1}$ are the probability distributions of $\mathcal{P}$ and $\mathcal{Q}$, respectively. The 1-Wasserstein distance is defined as

$$W_1(\mathcal{P}, \mathcal{Q}) = \inf_{\pi \in \Pi(\mathcal{P}, \mathcal{Q})} \sum_{j=0}^{N-1} \sum_{k=0}^{N-1} \pi_{jk} d(|\psi_j\rangle, |\psi_k\rangle), \tag{6}$$

where $\Pi(\mathcal{P}, \mathcal{Q})$ is the set of couplings $\pi$. Here, $\pi = (\pi_{jk})_{j,k=0}^{N-1}$ is a non-negative matrix satisfying $\sum_{j=0}^{N-1} \pi_{jk} = q_k$, $\sum_{k=0}^{N-1} \pi_{jk} = p_j$, $\pi_{jk} \geq 0$, and $d(|\psi_j\rangle, |\psi_k\rangle) = \frac{1}{2} \||\psi_j\rangle\langle\psi_j| - |\psi_k\rangle\langle\psi_k|\|_1 = \sqrt{1 - |\langle\psi_j|\psi_k\rangle|^2} \leq 1$, with $d(|\psi_j\rangle, |\psi_j\rangle) = 0$.

To bound $W_1(\mathcal{P}, \mathcal{Q})$, we construct an explicit coupling $\pi^* \in \Pi(\mathcal{P}, \mathcal{Q})$ with diagonal terms $\pi_{jj}^* = \min(p_j, q_j)$ and off-diagonal terms $\pi_{jk}^*$ (for $j \neq k$) distributing the remaining probability mass to satisfy the marginal constraints. The distance is then bounded as $W_1(\mathcal{P}, \mathcal{Q}) \leq \sum_{j=0}^{N-1} \sum_{k=0}^{N-1} \pi_{jk}^* d(|\psi_j\rangle, |\psi_k\rangle)$. Since $d(|\psi_j\rangle, |\psi_j\rangle) = 0$, only the off-diagonal terms contribute: $W_1(\mathcal{P}, \mathcal{Q}) \leq \sum_{j\neq k} \pi_{jk}^* d(|\psi_j\rangle, |\psi_k\rangle) \leq \sum_{j\neq k} \pi_{jk}^*$, because $d(|\psi_j\rangle, |\psi_k\rangle) \leq 1$. The sum of the off-diagonal terms is $\sum_{j\neq k} \pi_{jk}^* = \sum_{j=0}^{N-1} \sum_{k=0}^{N-1} \pi_{jk}^* - \sum_{j=0}^{N-1} \pi_{jj}^* = 1 - \sum_{j=0}^{N-1} \min(p_j, q_j) = \frac{1}{2} \sum_{j=0}^{N-1} |p_j - q_j| = \delta(p, q)$, since $\sum_{j,k} \pi_{jk}^* = \sum_j p_j = \sum_k q_k = 1$ and we use $\sum_{j=0}^{N-1} \min(p_j, q_j) = \sum_{j=0}^{N-1} \left( \frac{p_j + q_j - |p_j - q_j|}{2} \right) = 1 - \frac{1}{2} \sum_{j=0}^{N-1} |p_j - q_j|$. Thus, $W_1(\mathcal{P}, \mathcal{Q}) \leq \delta(p, q) \leq \frac{\varepsilon}{2}$, where the final inequality follows from Lemma 4.3.

For Lemma 4.4, where $p = q$, we have $W_1(\mathcal{P}, \mathcal{Q}) = 0$. This completes the proof. $\square$

Combining Lemmas 4.2 and 4.5, we bound the 1-Wasserstein distance between $\mathcal{Q}_t$ and the projected ensemble $\mathcal{P}$ as $W_1(\mathcal{Q}_t, \mathcal{P}) \leq W_1(\mathcal{Q}_t, \mathcal{Q}) + W_1(\mathcal{Q}, \mathcal{P}) \leq \frac{\varepsilon}{2} + \frac{\varepsilon}{2} = \varepsilon$. Using Lemma 4.4, the bound improves to: $W_1(\mathcal{Q}_t, \mathcal{P}) \leq \frac{\varepsilon}{2} + 0 = \frac{\varepsilon}{2}$. These results confirm that the MPE framework, supported by the construction of the unitary $V$ in Lemma 4.3, enables the parameterized distribution $\mathcal{P} = \mathcal{Q}_{\boldsymbol{\theta}}$ to approximate $\mathcal{Q}_t$, as required for the universality approximation theorem (Theorem 4.1).

## 5 LEARNING QUANTUM DISTRIBUTIONS WITH INCREMENTAL MPE

Theorem 4.1 guarantees that the MPE can approximate any target distribution $\mathcal{Q}_t$ over $n$-qubit pure states within a 1-Wasserstein distance of $\varepsilon > 0$. However, in this general case, constructing an efficient parameterized quantum distribution and collecting all states in the projected ensemble can require intensive resources. Equation 2 implies $n_a = \lceil \log N \rceil = \mathcal{O}(D \log(2/\varepsilon))$ ancilla qubits, which becomes inefficient when $D$ scales exponentially or polynomially (with high degree) with $n$. In practical scenarios with a structured $\mathcal{Q}_t$, we present an Incremental MPE framework that iteratively approximates $\mathcal{Q}_t$ through a layer-wise training scheme, reducing computational complexity and empirically improving the trainability and convergence for the optimization. We employ the fidelity-based 1-Wasserstein distance metric for training. The fidelity-based distance provides the upper-bound for the trace distance via the Fuchs-van de Graaf inequality: $d(|\mu\rangle, |\phi\rangle) \leq \sqrt{1 - \kappa(|\mu\rangle, |\phi\rangle)}$, ensuring that low 1-Wasserstein distance implies low $W_1$ (Equation 3.1).

### 5.1 INCREMENTAL MPE FRAMEWORK

Given a target distribution $\mathcal{Q}_t$, the learning process aims to construct a parameterized $\mathcal{Q}_{\boldsymbol{\theta}}$ that approximates $\mathcal{Q}_t$ through $K$ iterative cycles of unitary transformations and measurements.

First, we sample a training dataset $\mathcal{S} = \{|\psi_0\rangle, \ldots, |\psi_{N-1}\rangle\}$ consisting of $N$ pure $n$-qubit states drawn from $\mathcal{Q}_t$. The process continues with an initial ensemble $\tilde{\mathcal{S}}_0 = \{|\tilde{\psi}_j^{(0)}\rangle\}_j$, where the states are sampled from a random distribution, such as Haar product states. At each cycle $k = 0, \ldots, K-1$, we apply a parameterized unitary $V_k = V(\boldsymbol{\theta}_k)$ to the composite system of the data system $D$ (with $n_d = n$ qubits) and an auxiliary system $F$ (with $n_f$ qubits), initialized in the state $|\tilde{\psi}_j^{(k)}\rangle_D \otimes |0\rangle_F$. We can think $F$ represents the composite system of $A$ and $M\backslash D$. This is followed by a projective measurement on the ancilla system in the computational basis, yielding an outcome $\boldsymbol{z}_j^{(k)} \in \{0,1\}^{n_f}$ and a corresponding state $|\tilde{\psi}_j^{(k+1)}\rangle_D$ in the data system. The operation at cycle $k$ is formalized as:

$$\Phi_j^{(k)}(|\tilde{\psi}_j^{(k)}\rangle) = \frac{(I_D \otimes \Pi_F)V_k|\tilde{\Psi}_j^{(k)}\rangle}{\sqrt{\langle\tilde{\Psi}_j^{(k)}|V_k^\dagger(I_D \otimes \Pi_F)V_k|\tilde{\Psi}_j^{(k)}\rangle}} = |\tilde{\psi}_j^{(k+1)}\rangle_D \otimes |\boldsymbol{z}_j^{(k)}\rangle_F, \tag{7}$$

where $\Pi_F = |z_j^{(k)}\rangle\langle z_j^{(k)}|_F$ is the projector onto the ancilla measurement outcome, and $|\tilde{\Psi}_j^{(k)}\rangle = |\tilde{\psi}_j^{(k)}\rangle_D \otimes |0\rangle_F$. The resulting ensemble $\tilde{\mathcal{S}}_{k+1} = \{|\tilde{\psi}_j^{(k+1)}\rangle\}_j$ mirrors the structure of the MPE framework but is generated iteratively to reduce resource demands.

This process is repeated for $K$ cycles, with the sequence of parameterized unitaries $V_0, \ldots, V_{K-1}$ defining $\mathcal{Q}_{\boldsymbol{\theta}}$. The parameters $\boldsymbol{\theta}_k$ of $V_k$ are optimized to minimize a loss function $\mathcal{D}(\mathcal{S}, \tilde{\mathcal{S}}_{k+1})$, which measures the dissimilarity between the training dataset $\mathcal{S}$ and the ensemble $\tilde{\mathcal{S}}_{k+1}$. After optimization at the cycle $k$, $\boldsymbol{\theta}_k$ is fixed and the process optimizes $\boldsymbol{\theta}_{k+1}$ in the next cycle. This layer-wise training approach (Skolik et al., 2021; Zhang et al., 2024) decomposes the learning problem into $K$ manageable sub-tasks, each with a small number of trainable parameters, facilitating convergence.

In our numerical experiments, each $V_k$ is constructed using a Hardware Efficient Ansatz on $n_q = n_d + n_f$ qubits with $L$ layers as $V_k(\boldsymbol{\theta}_k) = \prod_{l=1}^{L} \tilde{\Omega}_k \tilde{W}_k(\boldsymbol{\theta}_k)$, where $\tilde{W}_k(\boldsymbol{\theta}_k) = \prod_{j=1}^{n_q} e^{-i\theta_{k,2j-1}\frac{X_j}{2}} e^{-i\theta_{k,2j-2}\frac{Y_j}{2}}$ and $\tilde{\Omega}_k = \prod_{j=1}^{n_q-1} CZ_{j,j+1}$. Here, $X_j$ and $Y_j$ are Pauli-X and Pauli-Y operators acting on the $j$-th qubit, implementing single-qubit rotations about the $y$- and $z$-axes, parameterized by $\theta_{k,2j-1}$ and $\theta_{k,2j-2}$, respectively. The $CZ_{a,b}$ gate is a two-qubit controlled-Z gate that applies a $Z$ operation to the target qubit (index $b$) when the control qubit (index $a$) is in the state $|1\rangle$, generating entanglement between qubit pairs in $\tilde{\Omega}_k$.

## 5.2 Metrics to compare ensembles

To quantify the similarity between ensembles, we employ a symmetric, positive definite quadratic kernel $\kappa(|\mu\rangle, |\phi\rangle)$ to define loss functions. This kernel can be computed efficiently using techniques such as the SWAP test (for state fidelity) or classical shadows (for classical-based computation) Huang et al. (2020). Our study employs three key metrics:

1. **Maximum Mean Discrepancy (MMD)**: The MMD distance between two ensembles $\mathcal{X} = \{|\mu_i\rangle\}_i$ and $\mathcal{Y} = \{|\psi_j\rangle\}_j$ is defined as:
$$\mathcal{D}_{\text{MMD}}(\mathcal{X}, \mathcal{Y}) = \bar{\kappa}(\mathcal{X}, \mathcal{X}) + \bar{\kappa}(\mathcal{Y}, \mathcal{Y}) - 2\bar{\kappa}(\mathcal{X}, \mathcal{Y}), \tag{8}$$
where $\bar{\kappa}(\mathcal{X}, \mathcal{Y}) = \mathbb{E}_{|\mu\rangle \in \mathcal{X}, |\phi\rangle \in \mathcal{Y}}[\kappa(|\mu\rangle, |\phi\rangle)]$.

2. **1-Wasserstein Distance**: Given a normalized kernel ($\kappa(|\phi\rangle, |\phi\rangle) = 1$ for all $|\phi\rangle$), we define a pairwise cost matrix $\mathbf{C} = (C_{i,j}) \in \mathbb{R}^{|\mathcal{X}| \times |\mathcal{Y}|}$ with $C_{i,j} = 1 - \kappa(|\mu_i\rangle, |\psi_j\rangle)$. The 1-Wasserstein distance is computed as the solution to the optimal transport problem:
$$\mathcal{D}_{\text{Wass}}(\mathcal{X}, \mathcal{Y}) = \min_{\mathbf{P}} \sum_{i,j} P_{i,j} C_{i,j}, \quad \text{s.t.} \quad \mathbf{P}\mathbf{1}_{|\mathcal{Y}|} = \mathbf{a}, \quad \mathbf{P}^\top \mathbf{1}_{|\mathcal{X}|} = \mathbf{b}, \quad \mathbf{P} \geq 0, \tag{9}$$
where $\mathbf{1}_{|\mathcal{X}|}$ and $\mathbf{1}_{|\mathcal{Y}|}$ are all-ones vectors of sizes $|\mathcal{X}|$ and $|\mathcal{Y}|$, respectively, and $\mathbf{a} \in \mathbb{R}^{|\mathcal{X}|}$, $\mathbf{b} \in \mathbb{R}^{|\mathcal{Y}|}$ are probability vectors (typically set to uniform, $\mathbf{a} = \frac{1}{|\mathcal{X}|}\mathbf{1}_{|\mathcal{X}|}, \mathbf{b} = \frac{1}{|\mathcal{Y}|}\mathbf{1}_{|\mathcal{Y}|}$).

3. **Vendi Score (VS)**: The Vendi Score (VS) (Friedman & Dieng, 2023) is a metric designed to evaluate the diversity of a set of samples. Given an ensemble $\mathcal{X} = \{|\mu_i\rangle\}_i$, and the normalized kernel matrix $\mathcal{K} = \kappa(|\mu_i\rangle, |\mu_j\rangle)$ defined in $\mathcal{X}$, the VS is computed as the exponential of the negative sum of the eigenvalues $\lambda_i$ (normalized by the sample size $N$) multiplied by their logarithms, or equivalently, the exponential of the negative trace of the normalized similarity matrix $\mathcal{K}/N$ times its logarithm. Mathematically, it is expressed as: $VS(\mathcal{X}) = \exp\left(-\sum_{i=1}^{N} \lambda_i \log \lambda_i\right) = \exp\left(-\text{tr}\left(\frac{\mathcal{K}}{N} \log \frac{\mathcal{K}}{N}\right)\right)$. The VS measures the spread or redundancy of samples: a high score indicates broad coverage with diverse and non-repetitive samples, while a low score suggests collapsed distributions.

# 6 Demonstration

To assess the efficacy of the Incremental MPE framework, we performed numerical experiments on two representative datasets: a synthetic clustered quantum state distribution and a quantum distribution derived from a chemistry dataset. Quantum circuits were simulated using the TensorCircuit library (Zhang et al., 2023), while JAX (Bradbury et al., 2018) facilitated automatic differentiation for gradient-based optimization. Circuit parameters were initialized uniformly in the interval $[-\pi, \pi]$, and optimization was conducted via the Adam algorithm with a learning rate of $0.001$.

## 6.1 MULTI-CLUSTER QUANTUM DISTRIBUTION

We consider a mixture of $n$-qubit pure states centered around distinct clusters, modeling multimodal quantum data relevant to applications such as quantum chemistry and error correction. The experiments demonstrate the framework's ability to approximate the target distribution $\mathcal{Q}_t$ with high fidelity, its scalability across varying qubit numbers, and its robustness to noise.

We construct a target distribution $\mathcal{Q}_t$ as a mixture of three clusters (40% with cluster 1, 40% with cluster 2, and 20% with cluster 3) of $n$-qubit ($n = 6$) pure states. Cluster 1 is centered on $|0\rangle^{\otimes n}$, cluster 2 on $|1\rangle^{\otimes n}$, and cluster 3 on the GHZ state $\frac{1}{\sqrt{2}}\left(|0\rangle^{\otimes n} + |1\rangle^{\otimes n}\right)$. For each cluster, noise is introduced by applying random single-qubit rotations with angles drawn from a Gaussian distribution ($\mathcal{N}(0, \sigma^2)$, $\sigma = 0.05$) to simulate quantum device imperfections.

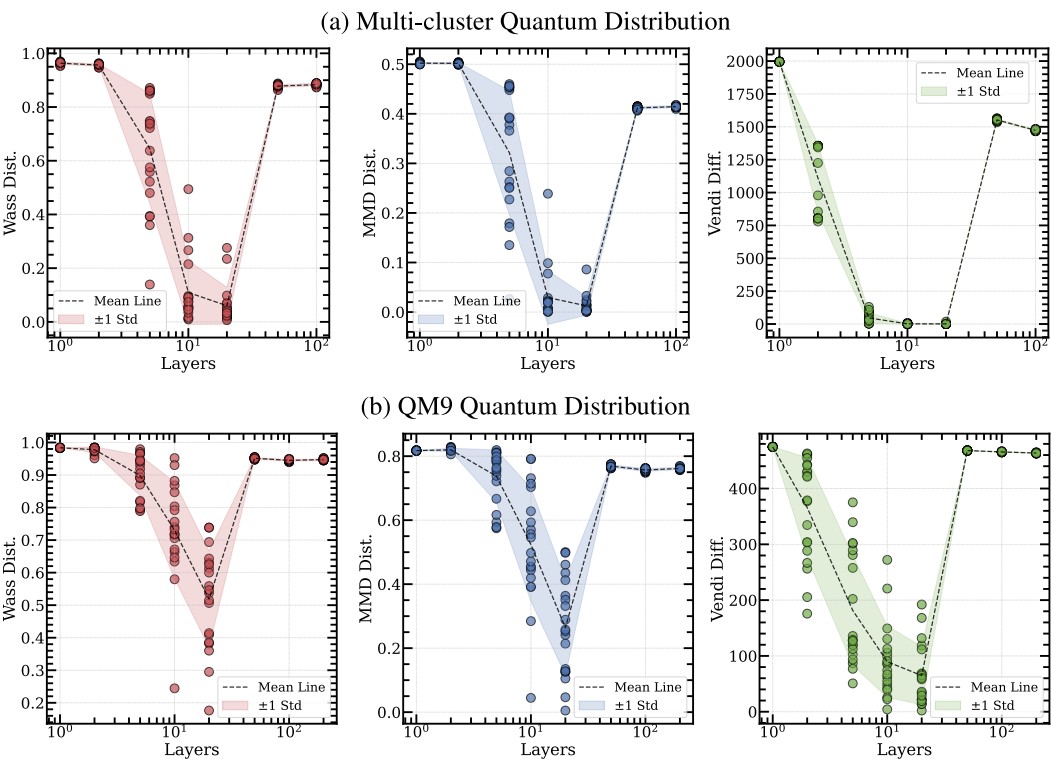

(a) Multi-cluster Quantum Distribution

(b) QM9 Quantum Distribution

Figure 2: Variation of the evaluation metric with changing circuit ansatz layers using the Incremental MPE framework to learn quantum distributions for (a) multi-cluster states and (b) molecular quantum states from a QM9 subset. Circle markers indicate individual trials, dotted lines show the mean over 20 trials, and shaded areas represent one standard deviation.

In our numerical experiments, we set $n_f = n/2$ and vary the number of incremental steps $K = 1, 2, 5, 10, 20, 50, 100$, with each unitary $V_k$ comprising $L = 100/K$ layers to maintain a constant total number of layers. Training to optimize each $V_k$ employs 1000 samples over $100 \times L$ epochs, with the number of epochs scaling with the number of layers, as more parameters necessitate additional epochs for effective optimization. We adopt mini-batch training, where each epoch consists of 10 iterations, processing 10 mini-batches of size $B = 100$. At each iteration, the loss function is the 1-Wasserstein distance $\mathcal{D}_{\text{Wass}}(S_{\text{train}}, S_{\text{out}})$, where $S_{\text{train}}$ is a set of $B$ quantum states sampled from the target distribution, and $S_{\text{out}}$ is a set of $B$ quantum states generated by the model.

Figure 2(a) depicts the variation of the evaluation metric with the number of layers, utilizing the 1-Wasserstein distance, MMD distance, and VS difference between 3000 generated and target samples. The experiments are conducted with 20 trials. All metrics indicate optimal performance around a specific number of layers $L$, where distances and differences are minimized, suggesting enhanced alignment between generated and target samples, though variability increases beyond this point. For a small $L$, even with a large number of steps $K = 100/L$, the expressivity of $V_k$ remains in-

sufficient, leading to local minima in each incremental step. Conversely, an excessively large $L$ introduces excessive expressivity in each $V_k$, leading to overparameterization or barren plateaus, resulting in optimization being challenging. The optimal $L$ achieves an average 1-Wasserstein distance of less than 0.1, an average MMD distance of less than 0.05, and an average VS difference of less than 1.0, facilitating the effective learning of the multi-cluster quantum distribution.

## 6.2 QM9 QUANTUM DISTRIBUTION

We demonstrate our framework to learn the QM9 dataset (Ramakrishnan et al., 2014), a widely recognized benchmark in computational chemistry. This dataset comprises approximately 134,000 small organic molecules, each with up to 9 heavy atoms (C, N, O, F) and additional hydrogens, totaling up to 29 atoms per molecule, along with their molecular properties and 3-D coordinates. Derived from the GDB-17 database (Ruddigkeit et al., 2012), QM9 is curated for quantum chemistry tasks, including molecular property prediction and 3-D structure generation. Given the current scale of our quantum simulation, evaluating the whole dataset is impractical. Therefore, we filter QM9 to include only molecules with exactly 7 heavy atoms and 2 distinct ring systems, yielding a specific subset of 488 molecules with uniform structural properties. Each 3-D molecule within this subset is encoded into a 7-qubit quantum state (see Appendix A.6 for details), enabling the task of learning the quantum data distribution corresponding to this QM9 subset.

In our numerical experiments, we set $n_d = 7$, $n_f = 3$, and vary the number of incremental steps $K = 1, 2, 5, 10, 20, 50, 100, 200$, with each unitary $V_k$ comprising $L = 200/K$ layers. We employ mini-batch training with a batch size $B = 100$ for each $V_k$, utilizing 200 training samples over $100 \times L$ epochs. The experiments are conducted over twenty trials. The loss function, measuring the divergence between two ensembles $S_{\text{train}}$ and $S_{\text{out}}$, is defined as a linear combination of the 1-Wasserstein distance and the Vendi Score (VS) square difference: $\mathcal{D}_{\text{Wass}}(S_{\text{train}}, S_{\text{out}}) + \lambda [VS(S_{\text{train}}) - VS(S_{\text{out}})]^2$, where $\lambda = 0.0001$ balances the contributions. Figure 2(b) illustrates the variation of the metrics with the number of layers, based on comparisons between 488 generated and target samples. The model exhibits optimal performance around $L = 20$ layers; however, the high distances indicate incomplete convergence. To potentially reduce the Wasserstein distance below 0.1, increasing the number of training epochs, $n_f$, and refining the ansatz circuit design could prove beneficial.

## 7 CONCLUSION

In this study, we have developed a universality approximation theorem for the MPE framework, demonstrating its ability to approximate any $n$-qubit pure state distribution within a specified 1-Wasserstein distance error. This theoretical result highlights the MPE framework's potential as a versatile tool for quantum data generation in QML. While the primary contribution lies in this universality theorem, the proposed Incremental MPE variant enhances practical applicability by mitigating optimization issues through layer-wise training, rendering it well-suited for NISQ devices. Numerical validations conducted on clustered quantum states and QM9 molecular datasets substantiate the framework's effectiveness in learning complex quantum distributions. These findings provide a robust foundation for advancing quantum generative modeling, with significant implications for quantum chemistry, materials science, and related fields.

**Limitations and Future Work.** While the universality approximation theorem presents the formal theoretical findings to approximate any distribution of quantum data with arbitrary error, the sample complexity often exhibits exponential scaling with the intrinsic dimension of the data manifold (Narayanan & Mitter, 2010). Future work may reduce this scale due to a specific shape of the target distribution, where smoother assumptions or symmetries can yield polynomial rates. The MPE framework relies on precise ancilla-assisted measurements, which may introduce errors in NISQ hardware due to imperfect gate operations. Additionally, while layer-wise training of Incremental MPE improves trainability, the computational cost of optimizing large-scale quantum circuits remains significant. Future work could focus on optimizing resource requirements for approximation protocols to enhance efficiency. Extending the MPE framework to mixed state distributions would broaden its applicability to open quantum systems. A potential direction is to incorporate the concept of the mixed projected ensemble (Yu et al., 2025), which is built from a local region of a quantum many-body system with a partial loss of measurement outcomes.

ETHICS STATEMENT

This research does not involve human subjects, sensitive datasets, or applications with direct societal harm. We have ensured compliance with the ICLR Code of Ethics, particularly regarding research integrity and transparency. All experiments were conducted using publicly available datasets, and no conflicts of interest or external sponsorship influenced the research outcomes.

REPRODUCIBILITY STATEMENT

To ensure the reproducibility of our results, we provide detailed information in the main paper, appendix, and supplementary materials. The proposed model and algorithms are described in the main text, with implementation details in Appendices 4.3 and A.6. Anonymous source code is available in the supplementary materials. For experiments, we used the QM9 dataset, with preprocessing steps fully documented in Appendix A.6.

LLM USAGE

We used a large language model (LLM) to assist with polishing the English language in this paper. Specifically, the LLM was used to enhance the grammar, clarity, and readability of the text without altering its original meaning. All LLM-generated content was carefully reviewed and edited by the authors to ensure accuracy, originality, and alignment with the research objectives. No LLMs were used for research ideation, data analysis, or generating discussion for the results.

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

# A APPENDIX

## A.1 BACKGROUND ON QUANTUM COMPUTING

We provide the essential concepts in quantum computing necessary for understanding our study. For a comprehensive treatment, we refer the reader to Nielsen & Chuang (2010).

**Quantum bit and quantum states**. A fundamental unit in quantum computing is the quantum bit, or qubit, which represents the state of a quantum system. Before measurement, a qubit can exist in a superposition of basis states, but upon measurement, it collapses into one of the basis states with probabilities determined by the quantum state.

The pure state of a quantum system consisting of $n$ qubits is described by a vector in a Hilbert space $\mathcal{H} = (\mathbb{C}^2)^{\otimes n}$. The mathematical representation of a quantum state depends on the choice of basis. For instance, using the orthogonal computational basis states $|0\rangle = \begin{pmatrix} 1 \\ 0 \end{pmatrix}$ and $|1\rangle = \begin{pmatrix} 0 \\ 1 \end{pmatrix}$, a single-qubit state can be expressed as a linear combination $|\psi\rangle = \alpha|0\rangle + \beta|1\rangle$, where $\alpha, \beta \in \mathbb{C}$ are complex amplitudes satisfying the normalization condition $|\alpha|^2 + |\beta|^2 = 1$. Computational basis states for multi-qubit systems are tensor products, such as $|01\rangle = |0\rangle \otimes |1\rangle = \begin{pmatrix} 0 \\ 1 \\ 0 \\ 0 \end{pmatrix}$. Any pure quantum state $|\psi\rangle \in \mathcal{H}$ satisfies $\langle\psi|\psi\rangle = 1$, where $\langle\psi|$ denotes the conjugate transpose of $|\psi\rangle$.

Pure states represent systems in definite quantum states, while mixed states describe statistical ensembles of pure states. A mixed state is represented by a density operator $\rho$, which is a positive semidefinite Hermitian operator with trace one ($\text{Tr}(\rho) = 1$). For a pure state $|\psi\rangle$, the density operator is $\rho = |\psi\rangle\langle\psi|$. For a mixed state as a probabilistic mixture of pure states $\{|\psi_i\rangle\}$ with probabilities $\{p_i\}$, the density operator is $\rho = \sum_i p_i |\psi_i\rangle\langle\psi_i|$. For example, the density matrix for $|0\rangle$ is $\rho_0 = |0\rangle\langle0| = \begin{pmatrix} 1 & 0 \\ 0 & 0 \end{pmatrix}$.

**Quantum gates and circuits**. A quantum computer operates via quantum circuits, consisting of wires (for qubits) and unitary gates that evolve quantum states. Each gate $U$ is a unitary operator on $\mathcal{H}$, and the circuit's overall action is the matrix product of these unitaries, computable via tensor products.

A fundamental single-qubit gate is the Hadamard gate $H$, which creates superposition from computational basis states: $H = \frac{1}{\sqrt{2}} \begin{pmatrix} 1 & 1 \\ 1 & -1 \end{pmatrix}$. Applying $H$ to $|0\rangle$ yields the equal superposition $\frac{1}{\sqrt{2}}(|0\rangle + |1\rangle)$, and to $|1\rangle$ yields $\frac{1}{\sqrt{2}}(|0\rangle - |1\rangle)$.

Examples of parameterized single-qubit gates include the rotation gates:

$$R_y(\theta) = \begin{pmatrix} \cos\left(\frac{\theta}{2}\right) & -\sin\left(\frac{\theta}{2}\right) \\ \sin\left(\frac{\theta}{2}\right) & \cos\left(\frac{\theta}{2}\right) \end{pmatrix}, \quad R_z(\theta) = \begin{pmatrix} e^{-i\theta/2} & 0 \\ 0 & e^{i\theta/2} \end{pmatrix}. \tag{10}$$

Common two-qubit gates include the controlled-Z (CZ) gate:

$$\text{CZ} = \begin{pmatrix} 1 & 0 & 0 & 0 \\ 0 & 1 & 0 & 0 \\ 0 & 0 & 1 & 0 \\ 0 & 0 & 0 & -1 \end{pmatrix}, \tag{11}$$

which applies a phase flip to the target qubit if the control qubit is $|1\rangle$.

An example of a parameterized two-qubit gate is the controlled rotation:

$$CR_x(\theta) = \begin{pmatrix} 1 & 0 & 0 & 0 \\ 0 & 1 & 0 & 0 \\ 0 & 0 & \cos\left(\frac{\theta}{2}\right) & -i\sin\left(\frac{\theta}{2}\right) \\ 0 & 0 & -i\sin\left(\frac{\theta}{2}\right) & \cos\left(\frac{\theta}{2}\right) \end{pmatrix}. \tag{12}$$

Another important parameterized two-qubit gate is the rotation around the ZZ axis:

$$R_{ZZ}(\theta) = \exp\left(-i\frac{\theta}{2}Z \otimes Z\right) = \begin{pmatrix} e^{-i\theta/2} & 0 & 0 & 0 \\ 0 & e^{i\theta/2} & 0 & 0 \\ 0 & 0 & e^{i\theta/2} & 0 \\ 0 & 0 & 0 & e^{-i\theta/2} \end{pmatrix}, \tag{13}$$

which generates entangling interactions between two qubits and is diagonal in the computational basis.

**Quantum measurement**. Quantum measurements extract classical information from a quantum state, fundamentally altering it via collapse. For a projective measurement defined by a Hermitian observable $M$ on $\mathcal{H}$, we decompose $M = \sum_m mP_m$, where $m$ are the distinct eigenvalues and $P_m$ are the corresponding orthogonal projectors satisfying $\sum_m P_m = I$ and $P_m P_{m'} = \delta_{mm'}P_m$. Given a pure state $|\psi\rangle$, the probability of outcome $m$ is $p(m) = \langle\psi|P_m|\psi\rangle = \langle\psi|P_m|\psi\rangle$. Upon observing $m$, the state collapses to the normalized post-measurement state $\frac{P_m|\psi\rangle}{\sqrt{p(m)}}$.

For mixed states described by $\rho$, the outcome probability generalizes to $p(m) = \text{Tr}(P_m\rho)$, and the updated density operator is $\rho' = \frac{P_m\rho P_m}{p(m)}$. In the common case of measuring individual qubits in the computational basis, the projectors are $P_0 = |0\rangle\langle0|$ and $P_1 = |1\rangle\langle1|$ per qubit, yielding binary outcomes with probabilities given by the diagonal elements of the reduced density matrix. This collapse from superposition to a definite basis state underscores the irreversible nature of measurement in quantum mechanics.

## A.2 LEARNING AN ENSEMBLE OF QUANTUM STATES

In quantum information science, ensembles of quantum states play a pivotal role in characterizing randomness and universality in quantum systems. An ensemble of quantum states $\mathcal{E} = \{(p_j, |\psi_j\rangle)\}$ consists of a set of pure quantum states $|\psi_i\rangle$ in a Hilbert space $\mathcal{H}$, each weighted by a probability $p_j$ such that $\sum_j p_j = 1$. This ensemble represents a probabilistic mixture, capturing stochastic processes that generate quantum states. Unlike an individual wave function or density matrix, randomness is an emergent property of the ensemble.

We want to clarify that learning an ensemble of quantum states is a different problem from preparing the density matrix $\rho = \sum_j p_j|\psi_j\rangle\langle\psi_j|$ of the ensemble. Here, $\rho$ alone does not uniquely determine the ensemble, as multiple distinct ensembles can yield the same density matrix. The goal of learning an ensemble of quantum states is to learn how to sample quantum states from an unknown distribution. The density matrix encodes the average expectation values of an observable $O$:

$$\sum_j p_j\langle\psi_j|O|\psi_j\rangle = \text{Tr}(O\rho). \tag{14}$$

The density matrix is insufficient to distinguish ensembles, which require higher-order moments. For example, the $k$-th moment operator is $\rho_{\mathcal{E}}^{(k)} = \sum_j p_j\left(|\psi_j\rangle\langle\psi_j|\right)^{\otimes k}$. This acts on $k$ copies of the Hilbert space and describes an incoherent sum of $k$ identical states. From the $k$-th moment operator, we can calculate the $k$-the moment of the observable $O$ as:

$$\sum_j p_j\left(\langle\psi_j|O|\psi_j\rangle\right)^k = \text{Tr}(O^{\otimes k}\rho_{\mathcal{E}}^{(k)}). \tag{15}$$

In general, learning an ensemble involves estimating its density matrix, moments, or nonlinear properties from samples. While the first moment (density matrix) is accessible via standard quantum state tomography, higher moments require statistical estimation from labeled samples drawn from $\mathcal{E}$. One of the most interesting problems is to construct an ensemble that mimics (or approximates) the Haar ensemble up to the $k$-th moment. An $\varepsilon$-approximate $k$-designs satisfies $\|\rho_{\mathcal{E}}^{(k)} - \rho_{\text{Haar}}^{(k)}\| \leq \varepsilon$. Approximate designs emerge in physical systems, such as random unitary circuits or Hamiltonian evolutions.

Many projected ensembles (MPEs) offer a natural realization of $\varepsilon$-approximate $k$-designs (Choi et al., 2023; Cotler et al., 2023). Given a many-body state $|\Psi\rangle$ on subsystems $A$ ($n_a$ qubits) and $M$ ($n_m$ qubits), projective measurements on $A$ in a local basis $\{|z_A\rangle\}$ yield the ensemble on $M$ as

$$\mathcal{E}_{\Psi,M} = \{(p(z_A), |\psi(z_A)\rangle_M)\}_{z_A}, \tag{16}$$

with $p(z_A) = \langle\Psi|\,(|z_A\rangle\langle z_A| \otimes \mathbf{1}_M)\,|\Psi\rangle$ and $|\psi(z_A)\rangle_M = (\langle z_A| \otimes \mathbf{1}_M)\,|\Psi\rangle/\sqrt{p(z_A)}$. The MPE generates a classical probability distribution $p(z_A)$ over measurement outcomes, but the projected states $|\psi(z_A)\rangle_M$ are genuinely quantum. Such ensembles approximate $k$-designs for generic generator states $|\Psi\rangle$, particularly in chaotic systems when $n_a$ is sufficiently large.

MPEs were originally developed to approximate $k$-designs from Haar-random ensembles. However, in our study, they are used innovatively to prove the universality of learning distributions from quantum states. While low-entanglement MPEs may be classically simulable, the framework's advantage lies in quantum hardware efficiently preparing the many-body state $|\Psi\rangle$ for sampling complex, non-local distributions (e.g., in quantum chemistry, where classical methods struggle with superposition and entanglement). This offers the potential for quantum speedups over classical generative models for tasks such as simulating molecular ensembles, as classical sampling from such distributions can require exponential resources.

### A.3 Challenges in Quantum Machine Learning (QML)

QML has garnered significant interest for its potential to leverage quantum systems for enhanced data processing and generative tasks. However, recent advancements have highlighted several challenges and negative results that temper claims of quantum advantage. This appendix provides an overview of these issues, drawing from key literature, and positions our MPE framework in this context. We focus on dequantization results, trainability bottlenecks such as barren plateaus, and classical simulability, while noting promising mitigation strategies.

**Dequantization and Classical Equivalents.** A growing body of work demonstrates that many QML algorithms, initially thought to offer exponential speedups, can be "dequantized"—simulated classically with comparable efficiency under realistic assumptions, such as access to classical data models (e.g., length-squared sampling). For instance, Tang (2019) provides dequantization for quantum recommendation systems, showing that classical algorithms can achieve similar performance with polynomial resources. Tang (2021) extends this to quantum principal component analysis, providing a classical counterpart that matches quantum outputs under standard data access. Chia et al. (2020) and Gharibian & Le Gall (2022) dequantize aspects of quantum singular value transformations (Gilyén et al., 2019), revealing classical simulations for tasks like quantum linear algebra.

More recent studies, such as Cerezo et al. (2025) and Gil-Fuster et al. (2025), apply dequantization to variational QML models, underscoring that claimed advantages often vanish when considering trivial datasets or sampling models of trivial distributions. These results suggest that QML's edge may be limited to regimes with inherent quantum structure, such as high-entanglement quantum many-body systems, where classical simulation becomes inefficient. In our MPE framework, we position it as potentially resistant to such dequantization in these high-entanglement scenarios. MPE leverages measurement-induced ensembles from many-body wave functions, which can capture non-local quantum correlations that classical methods struggle to replicate efficiently. While we do not claim a guaranteed quantum advantage, this motivates further exploration of MPE for tasks like quantum chemistry simulations, where entanglement plays a central role.

**Barren Plateaus.** A major bottleneck in QML using PQCs is the barren plateau phenomenon, where gradients vanish exponentially with system size or circuit depth, rendering optimization intractable (McClean et al., 2018). This arises from random initialization in high-dimensional parameter spaces, global measurements, excessive entanglement in the initial state, the circuit ansatz, and the existence of hardware noise, leading to flat loss landscapes (Larocca et al., 2025). Our Incremental MPE variant aims to address this by leveraging layer-wise training (Skolik et al., 2021), but full theoretical guarantees for this heuristic are lacking.

**Backpropagation Scaling.** The backpropagation scaling problem in PQCs refers to the inefficiency of gradient computation during the training of variational quantum algorithms. Unlike classical

neural networks, where gradients scale with constant or logarithmic overhead via backpropagation, PQCs rely on stochastic measurements and methods like the parameter-shift rule, leading to costs that grow linearly with the number of parameters. This arises from requiring separate circuit evaluations per parameter, amplified by shot noise, making large-scale optimization impractical. There are promising directions to mitigate this issue, such as constructing structured PQCs with commuting generators to enable parallel gradient estimation (Bowles et al., 2025), balancing expressivity and trainability (Chinzei et al., 2025), and classical surrogates for loss functions where classical approximations guide quantum optimization (Recio-Armengol et al., 2025).

These challenges could limit the practicality and trainability of MPE in real quantum hardware. However, the argument for classical simulability in training is particularly relevant for generative models: one can classically train a variational state by minimizing expectation values, but sampling from such states classically is often prohibitively expensive. This highlights a key distinction: while training may be dequantized, quantum hardware offers advantages for sampling entangled ensembles. From this perspective, our universality theorem is complementary. It ensures MPE can, in principle, capture any pure-state distribution before optimizing for hardware, focusing on expressivity rather than guaranteed speedup.

### A.4    PROOF FOR THE BOUND OF $\delta$-COVERING NUMBER

We present an intuition for an iterative algorithm to construct a $\delta$-net of a given quantum distribution $\mathcal{Q}_t$. Pick a point $|\psi_1\rangle \sim \mathcal{Q}_t$ arbitrarily sampled from $\mathcal{Q}_t$, then pick $|\psi_2\rangle \sim \mathcal{Q}_t$ that is farther than $\delta$ from $|\psi_1\rangle$, then pick $|\psi_3\rangle \sim \mathcal{Q}_t$ that is farther than from both $|\psi_1\rangle$ and $|\psi_2\rangle$, and so on. If $\mathcal{Q}_t$ is compact, this process stops in finite time and gives an $\delta$-net of $\mathcal{Q}_t$.

In quantum mechanics, the Hilbert space $\mathbb{C}^D$ (a $D$-dimensional complex vector space) describes a quantum system with $D$ possible basis states. A pure state of this system is represented by a unit vector $|\psi\rangle \in \mathbb{C}^D$ satisfying $\langle\psi|\psi\rangle = 1$, but physically equivalent states differ by a global phase: $|\psi\rangle \sim e^{i\theta}|\psi\rangle$ for any real $\theta$, as this phase does not affect observable quantities like probabilities or expectation values. Thus, the set of distinct pure states is not the full unit sphere $S^{2D-1} \subset \mathbb{R}^{2D}$ (real/imaginary parts of $\mathbb{C}^D$), but rather the quotient space obtained by identifying vectors that differ by a phase factor from the group $U(1)$ (unit complex numbers). This quotient is the complex projective space $\mathbb{CP}^{D-1}$, defined as $\mathbb{CP}^{D-1} = S^{2D-1}/U(1)$, where each point in $\mathbb{CP}^{D-1}$ corresponds to a 1-dimensional complex subspace (a "ray") in $\mathbb{C}^D$. Then if we define the manifold $\mathcal{M} = \{|\psi\rangle\langle\psi| : |\psi\rangle \in \mathbb{C}^D\}$ of pure states in $\mathbb{C}^D$, we have the relation $\mathcal{M} \cong \mathbb{CP}^{D-1}$.

Let $S \subset \mathbb{C}^{2^n}$ be a fixed $D$-dimensional complex subspace (i.e., $S$ has an orthonormal basis with $D$ elements $|e_1\rangle, \ldots, |e_D\rangle$), and define $\mathcal{K} = \{|\psi\rangle\langle\psi| : |\psi\rangle \in S, \langle\psi|\psi\rangle = 1\}$ as the submanifold of pure states supported entirely within $S$. Equipped with the trace distance $d(\rho, \sigma) = \frac{1}{2}\|\rho - \sigma\|_1$, the metric space $(\mathcal{K}, d)$ is isometric to the manifold of pure states on $\mathbb{C}^D$ under the induced Fubini-Study metric (rescaled to match trace distance). This ensures that covering numbers, volumes, and discretization strategies for $K$ inherit directly from $\mathbb{CP}^{D-1}$, without dilution from the larger space.

We consider the $\delta$-net discretizing the manifold $\mathcal{M}$ of pure states in $\mathbb{C}^D$ under the trace distance metric $d$. Based on Lemma 1 in Akibue et al. (2022), we estimate $\mathcal{N}(\mathcal{M}, d, \delta)$—the cardinality of the smallest $\delta$-net of $\mathcal{M}$ and derive formal lower and upper bounds in terms of $D$ and $\delta$.

**Lemma A.1** (Bound for covering number). *Let $\mathcal{M} = \{|\psi\rangle\langle\psi| : |\psi\rangle \in \mathbb{C}^D\}$ be the manifold of pure states in $\mathbb{C}^D$, equipped with the trace distance $d$. For $\delta \in (0, 1]$ and $D \geq 2$:*

$$(1/\delta)^{2(D-1)} \leq \mathcal{N}(\mathcal{M}, d, \delta) \leq 5 \cdot D \ln(D) \cdot (1/\delta)^{2(D-1)} \tag{17}$$

*Proof.* The proof relies on volumetric arguments using the unitarily invariant probability measure $\mu$ on $\mathcal{P}(\mathbb{C}^D)$, which normalizes the total volume to $\mu(\mathcal{P}(\mathbb{C}^D)) = 1$. The measure $\mu$ is a probability measure on $\mathbb{CP}^{D-1}$, meaning it assigns sizes to subsets such that the entire space has measure 1. It is derived from the Haar measure on the unitary group $U(D)$, inducing a uniform distribution: picking a random unitary and applying it to a fixed state (e.g., $|0\rangle$) samples uniformly from $\mu$. Without normalization, the raw volume of $\mathbb{CP}^{d-1}$ under Fubini-Study is $\pi^{D-1}/(D-1)!$ but we scale it to 1 for convenience to convert absolute volumes into probabilities.

**Proof for the lower bound**. First, we use the result presented in Appendix A in Akibue et al. (2022) to derive the volume of $\delta$-ball $B_\delta(\phi) := \{\psi \in \mathcal{P}(\mathbb{C}^D) : d(\phi, \psi) < \delta\}$ as follows:

$$\forall D \in \mathbb{N}, \forall \delta \in (0, 1], \forall \phi \in \mathcal{P}(\mathbb{C}^D), \quad \mu(B_\delta(\phi)) = \delta^{2(D-1)}. \tag{18}$$

Here, for the convenient with $|\psi\rangle, |\phi\rangle \in \mathbb{C}^D$, we write $\psi = |\psi\rangle\langle\psi| \in \mathcal{P}(\mathbb{C}^D)$ and $\phi = |\phi\rangle\langle\phi| \in \mathcal{P}(\mathbb{C}^D)$, and the trace distance $d(\phi, \psi) = \frac{1}{2}\|\phi - \psi\|_1$.

Since the total measure is 1 (as normalized) and each ball $B_\delta(\phi)$ has measure $\delta^{2(D-1)}$, to cover the space at least $1/\delta^{2(d-1)}$ balls are needed. Formally,

$$\mathcal{N}(\mathcal{M}, d, \delta) \geq \frac{\mu(\mathcal{P}(\mathbb{C}^D))}{\max_\phi \mu(B_\delta(\phi))} = \frac{1}{\delta^{2(D-1)}}. \tag{19}$$

**Proof for the upper bound**. For the upper bound, based on Appendix B in Akibue et al. (2022), we construct an explicit $\delta$-net using a greedy probabilistic method. The idea is to sample random pure states to cover most of the space, then greedily add points to cover the remainder. Let $D_{\text{eff}} = 2(D - 1) \geq 2$ (effective real dimension of the projective space). Sample $J_R$ random pure states $\{\phi_j\}_{j=0}^{J_R-1}$ from $\mu^{J_R}$. The expected uncovered measure in the region $(A^c)$ not covered by $A := \cup_{j=0}^{J_R-1} B_{\delta_R}(\phi_j)$ is calculated as follows:

$$\int d\mu^{J_R} \mu(A^C) = \int d\mu^{J_R} \int d\mu(\psi) \prod_{j=1}^{J_R} I[d(\psi, \phi_j) \geq \delta_R] \tag{20}$$

$$= \int d\mu(\psi) \prod_{j=1}^{J_R} \int d\mu(\phi_j) I[d(\psi, \phi_j) \geq \delta_R] \tag{21}$$

$$\leq \left(1 - \delta_R^{D_{\text{eff}}}\right)^{J_R} \leq \exp(-J_R \delta_R^{D_{\text{eff}}}), \tag{22}$$

where we use Fubini's theorem and $\mu(B_\delta(\phi)) = \delta^{D_{\text{eff}}}$. Here, $I[X] \in \{0, 1\}$ is the indicator function, i.e., $I[X] = 1$ iff $X$ is true. Thus, there exists a set $\{\phi_j\}_{j=0}^{J_R-1}$ with $\mu(A^c) \leq \exp(-J_R \delta^{D_{\text{eff}}})$. Now, pack disjoint $\delta_P$-balls ($\delta_P \leq \delta_R \leq 1$) into $A^c$ with centers $\{\psi_j\}_{j=1}^{J_P}$ as much as possible (greedy packing). The packing gives the following estimation:

$$J_P \leq \frac{\mu(A^c)}{\delta_P^{D_{\text{eff}}}} \leq \frac{\exp(-J_R \delta_R^{D_{\text{eff}}})}{\delta_P^{D_{\text{eff}}}}. \tag{23}$$

The combined set $\{\phi_j\}_{j=0}^{J_R-1} \cup \{\psi_j\}_{j=0}^{J_P-1}$ covers with radius $\delta_R + \delta_P = \delta$ and size $J = J_R + J_P$. Set $J_R = \lceil D_{\text{eff}} \delta_R^{-D_{\text{eff}}} \ln(\delta_R/\delta_P) \rceil$, $\delta_P = \delta_R/x$, $\delta_R = x\delta/(1 + x)$ with $x \geq 1$. This yields:

$$J = J_R + J_P \leq \left\lceil \frac{D_{\text{eff}} \ln x}{\delta_R^{D_{\text{eff}}}} \right\rceil + \frac{1}{\delta_R^{D_{\text{eff}}}} \leq \frac{1}{\delta^{D_{\text{eff}}}} \left\{ \left(1 + \frac{1}{x}\right)^{D_{\text{eff}}} (D_{\text{eff}} \ln x + 1) + 1 \right\} = \frac{\alpha(D_{\text{eff}}, x)}{\delta^{D_{\text{eff}}}}, \tag{24}$$

where $\alpha(D_{\text{eff}}, x) = (1 + 1/x)^{D_{\text{eff}}} (D_{\text{eff}} \ln x + 1) + 1$.

Now, we select $x = D_{\text{eff}} \ln D_{\text{eff}} > 1$ and consider the function

$$f(D) = \frac{\alpha(D_{\text{eff}}, D_{\text{eff}} \ln D_{\text{eff}})}{D \ln D}. \tag{25}$$

Numerically, we can check that $\frac{\partial f}{\partial D}(D = 2) > 0$, $\frac{\partial f}{\partial D}(D = 3) > 0$, and $\frac{\partial f}{\partial D}(D_0) < 0$ for $D_0 \geq 4$. If we think $D$ is a continuous real variable, then the derivative $\frac{\partial f}{\partial D}$ has a critical point $D^* \in (3, 4)$. Numerical calculation provides that $D^* \approx 3.032879$ and $f(D^*) \approx 4.927605 < 5$. Therefore, we have the following upper bound for $\mathcal{N}(\mathcal{M}, d, \delta)$:

$$\mathcal{N}(\mathcal{M}, d, \delta) \leq J \leq \frac{D \ln D}{\delta^{D_{\text{eff}}}} f(D) \leq \frac{D \ln D}{\delta^{D_{\text{eff}}}} f(D^*) < \frac{5D \ln D}{\delta^{D_{\text{eff}}}} = 5 \cdot D \ln(D) \cdot (1/\delta)^{2(D-1)}. \tag{26}$$

$\square$

Since $\mathcal{Q}_t \subseteq \mathcal{K}$, then we obtain $\mathcal{N}(\mathcal{Q}_t, d, \delta) \leq \mathcal{N}(\mathcal{K}, d, \delta) = \mathcal{N}(\mathcal{M}, d, \delta)$, which is the upper bound in Equation 2.

### A.5 PROOF OF LEMMA 4.3

The proof constructs a unitary $V$ acting on the ancilla system $A$ (with $n_a$ qubits) and the hidden system $M$ (with $n_m = n_a + \lceil \log_2(1/\varepsilon) \rceil$ qubits) to generate a probability distribution $p$ that approximates the target distribution $q$. Following Kurkin et al. (2025), we provide a concrete construction using an Instantaneous Quantum Polynomial (IQP) (Shepherd & Bremner, 2009; Bremner et al., 2010) circuit architecture. The proof proceeds in three steps: defining the IQP circuit, generating a logical model state, and approximating the target probability distribution.

#### A.5.1 STEP 1: IQP CIRCUIT ARCHITECTURE

IQP circuits form a class of quantum circuits consisting of commuting gates that are diagonal in the $Z$ basis. A parameterized IQP circuit on $n$ qubits consists of three components: (1) Hadamard gates $H^{\otimes n}$ on all qubits (initialized at $|0\rangle^{\otimes n}$) to create a uniform superposition, (2) a layer of parameterized gates of the form $\exp(i\theta_j Z_{\boldsymbol{g}_j})$, where $Z_{\boldsymbol{g}_j}$ is a tensor product of Pauli $Z$ operators acting on a subset of qubits specified by the nonzero entries of $\boldsymbol{g}_j \in \{0,1\}^n$, and (3) another layer of Hadamard gates $H^{\otimes n}$. Formally, an IQP circuit is $U = H^{\otimes n} D H^{\otimes n}$, where $D = \exp\left(i \sum_j \theta_j Z_{\boldsymbol{g}_j}\right)$. A parameterized IQP circuit with hidden units is a parameterized IQP circuit in which a chosen subset of qubits is traced out.

IQP circuits are particularly useful for sampling problems and exhibit properties that make them hard to simulate classically under certain complexity assumptions. In our framework, parameterized IQP circuits with hidden units provide an efficient parameterization for generating complex probability distributions over measurement outcomes.

To describe the parameterized IQP circuits with hidden units, we initialize the system in the state $|0\rangle^{\otimes(n_a+n_m)}$ and apply the unitary $V$ as:

1. **First Layer**: Apply Hadamard gates to all qubits, $H^{\otimes(n_a+n_m)}$, creating a uniform superposition:

$$|0\rangle^{\otimes(n_a+n_m)} \to \frac{1}{\sqrt{2^{n_a+n_m}}} \sum_{\boldsymbol{j}\in\{0,1\}^{n_a}} \sum_{\boldsymbol{k}\in\{0,1\}^{n_m}} |\boldsymbol{j}\rangle_A |\boldsymbol{k}\rangle_M. \tag{27}$$

2. **Middle Layer**: Apply a parameterized diagonal gate $D(\boldsymbol{\theta}) = \prod_{\boldsymbol{j}\in\{0,1\}^{n_a}, \boldsymbol{k}\in\{0,1\}^{n_m}} e^{i\theta_{\boldsymbol{j},\boldsymbol{k}} Z_{\boldsymbol{j},\boldsymbol{k}}}$, where $Z_{\boldsymbol{j},\boldsymbol{k}}$ is a tensor product of Pauli-$Z$ operators acting on subsets of qubits in $A \otimes M$, and $\theta_{\boldsymbol{j},\boldsymbol{k}} \in \mathbb{R}$ are trainable phases encoding the target distribution. The resulting state is:

$$|\psi\rangle = \frac{1}{\sqrt{2^{n_a+n_m}}} \sum_{\boldsymbol{j}\in\{0,1\}^{n_a}} \sum_{\boldsymbol{k}\in\{0,1\}^{n_m}} e^{i\theta_{\boldsymbol{j},\boldsymbol{k}}} |\boldsymbol{j}\rangle_A |\boldsymbol{k}\rangle_M. \tag{28}$$

   This is referred to as the Uniform Mixture Approximation (UMA) state.

3. **Final Layer**: Apply Hadamard gates ($H^{\otimes n_a} \otimes I^{\otimes n_m}$) to the ancilla system to prepare the state for measurement in the computational basis of $A$.

#### A.5.2 STEP 2: GENERATING THE LOGICAL MODEL STATE

To approximate the target distribution $q = \{q_b\}_{b=0,1,\ldots,2^{n_a}-1}$, we consider a logical model state defined by a mapping $v : \{0, 1, \ldots, 2^{n_m} - 1\} \to \{0, 1, \ldots, 2^{n_a} - 1\}$, where $n_m > n_a$. Here, we use bold letters for binary index, such as $\boldsymbol{b} \in \{0,1\}^{n_a}$ and normal letters for their decimal equivalent, such as $b \in \{0, 1, \ldots, 2^{n_a} - 1\}$. Therefore, $v(\boldsymbol{b})$ has the same meaning with $v(b)$. We define the state:

$$|\psi'\rangle = \frac{1}{\sqrt{2^{n_m}}} \sum_{k=0}^{2^{n_m}-1} |v(k)\rangle_A |k\rangle_M, \tag{29}$$

where $v(k)$ maps the hidden state index $k$ to an ancilla state index. Measuring the ancilla system $A$ in the computational basis yields outcome $\boldsymbol{b} \in \{0,1\}^{n_a}$ with probability:

$$p(\boldsymbol{b}) = \frac{|\{\boldsymbol{k} \in \{0,1\}^{n_m} : v(\boldsymbol{k}) = \boldsymbol{b}\}|}{2^{n_m}}. \tag{30}$$

We show that the UMA state $|\psi\rangle$ can be transformed into $|\psi'\rangle$ via a unitary, and that applying $H^{\otimes n_a} \otimes I^{\otimes n_m}$ to $|\psi'\rangle$ produces:

$$(H^{\otimes n_a} \otimes I^{\otimes n_m})|\psi'\rangle = \frac{1}{\sqrt{2^{n_a+n_m}}} \sum_{\boldsymbol{j} \in \{0,1\}^{n_a}} \sum_{\boldsymbol{k} \in \{0,1\}^{n_m}} (-1)^{v(\boldsymbol{k})\cdot\boldsymbol{j}} |\boldsymbol{j}\rangle_A |\boldsymbol{k}\rangle_M, \qquad (31)$$

where $v(\boldsymbol{k}) \cdot \boldsymbol{j}$ is the inner product modulo 2. This state is a UMA state with phases $\theta_{\boldsymbol{j},\boldsymbol{k}} = 0$ if $v(\boldsymbol{k}) \cdot \boldsymbol{j}$ is even, and $\theta_{\boldsymbol{j},\boldsymbol{k}} = \pi$ if $v(\boldsymbol{k}) \cdot \boldsymbol{j}$ is odd. Applying $H^{\otimes n_a} \otimes I^{\otimes n_m}$ to this state recovers $|\psi'\rangle$, and measuring $A$ in the computational basis yields the same probability distribution $p(\boldsymbol{b})$.

### A.5.3 STEP 3: PROBABILITY APPROXIMATION

The probability of outcome $\boldsymbol{b} \in \{0,1\}^{n_a}$ is:

$$p(\boldsymbol{b}) = \frac{c_{\boldsymbol{b}}}{2^{n_m}}, \quad \text{where} \quad c_{\boldsymbol{b}} = |\{\boldsymbol{k} \in \{0,1\}^{n_m} : v(\boldsymbol{k}) = \boldsymbol{b}\}|, \qquad (32)$$

and $c_{\boldsymbol{b}}$ is the number of hidden states mapped to outcome $\boldsymbol{b}$. To ensure $p(\boldsymbol{b}) \approx q(\boldsymbol{b})$, we choose the mapping $v$ as follows:

1. For each $\boldsymbol{b} \in \{0,1\}^{n_a}$, set $c_{\boldsymbol{b}} = \lfloor q(\boldsymbol{b})2^{n_m} \rfloor$. Compute the sum $S = \sum_{\boldsymbol{b} \in \{0,1\}^{n_a}} c_{\boldsymbol{b}} \leq 2^{n_m}$, with $2^{n_m} - S \leq 2^{n_a}$.

2. If $S < 2^{n_m}$, distribute the remaining $2^{n_m} - S$ states by incrementing $c_{\boldsymbol{b}} = \lfloor q(\boldsymbol{b})2^{n_m} \rfloor + 1$ for the first $2^{n_m} - S$ outcomes $\boldsymbol{b}$, ensuring $\sum_{\boldsymbol{b} \in \{0,1\}^{n_a}} c_{\boldsymbol{b}} = 2^{n_m}$.

3. Assign hidden states $k = 0, 1, \ldots, 2^{n_m} - 1$:
   - Assign the first $c_{\boldsymbol{b}_1}$ states ($k = 0, \ldots, c_{\boldsymbol{b}_1} - 1$) to $v(k) = \boldsymbol{b}_1$ (e.g., $\boldsymbol{b}_1 = 00\ldots0$).
   - Assign the next $c_{\boldsymbol{b}_2}$ states ($k = c_{\boldsymbol{b}_1}, \ldots, c_{\boldsymbol{b}_1} + c_{\boldsymbol{b}_2} - 1$) to $v(k) = \boldsymbol{b}_2$, and continue until all $2^{n_m}$ states are assigned.

The error for each outcome is:

$$|q(\boldsymbol{b}) - p(\boldsymbol{b})| = \left| q(\boldsymbol{b}) - \frac{c_{\boldsymbol{b}}}{2^{n_m}} \right| = \frac{|q(\boldsymbol{b})2^{n_m} - c_{\boldsymbol{b}}|}{2^{n_m}} \leq \frac{1}{2^{n_m}}. \qquad (33)$$

The total variation distance is:

$$\delta(p,q) = \frac{1}{2} \sum_{\boldsymbol{b} \in \{0,1\}^{n_a}} |p(\boldsymbol{b}) - q(\boldsymbol{b})| \leq \frac{1}{2} \cdot 2^{n_a} \cdot \frac{1}{2^{n_m}} = \frac{2^{n_a}}{2^{n_m+1}} = \frac{1}{2^{n_m-n_a+1}}. \qquad (34)$$

Choosing $n_m = n_a + \lceil \log_2(1/\varepsilon) \rceil$, we have: $2^{n_m-n_a} \geq \frac{1}{\varepsilon} \implies \frac{1}{2^{n_m-n_a}} \leq \varepsilon$. Thus: $\delta(p,q) \leq \frac{1}{2} \cdot \frac{1}{2^{n_m-n_a}} \leq \frac{\varepsilon}{2}$. This completes the proof, with the unitary $V = (H^{\otimes n_a} \otimes I^{\otimes n_m}) \cdot D(\boldsymbol{\theta}) \cdot H^{\otimes(n_a+n_m)}$ explicitly constructed to achieve the desired approximation.

### A.6 ENCODING 3-D MOLECULES TO QUANTUM STATES

This process ensures compatibility with quantum amplitude encoding, which requires input vectors to be normalized to unit norm. The molecules in the QM9 dataset—small organic compounds with up to 9 heavy atoms (C, N, O, F) plus hydrogens, totaling up to 29 atoms per molecule—are represented as attributed point clouds. Each molecule (index $j = 0, 1, \ldots, N - 1$ in the dataset) is denoted as $\{(\boldsymbol{v}_i^j, \boldsymbol{a}_i^j)\}_{i=0}^{m_j-1}$, where $m_j$ is the number of atoms, $\boldsymbol{v}_i^j \in \mathbb{R}^3$ are the 3-D coordinates, and $\boldsymbol{a}_i^j \in \{0,1\}^k$ is the one-hot encoded atom type, corresponding to $j$th molecule. Here, for simplification, we only consider heavy atoms in our model, leading to the number of atom types $k = 4$. The dataset can be represented by $[\{(\boldsymbol{v}_i^j, \boldsymbol{a}_i^j)\}_{i=0}^{m_j-1}]_{j=0}^{M-1}$.

The encoding needs to address challenges of arbitrary translations, rotations, atom ordering, and quantum normalization constraints, where the amplitudes of encoded quantum states are non-negative real values to simplify the encoding and reconstruction process. We adopt the encoding method in Rathi et al. (2023); Wu et al. (2024) with details in the following steps.

1. **Structural normalization for unique representation:** Atoms are reordered using canonical SMILES strings generated via RDKit toolkit (rdk), ensuring a consistent, graph-based ordering independent of the original input.

2. **Conformation fixing (translation and rotation):** The molecule is centered by subtracting the centroid (center of mass) from all coordinates, aligning it to the origin. It then rotates the position of the first atom in the SMILES string onto the z-axis.

3. **Positive octant adjustment:** After centering and rotation, coordinates may include negative values. The minimum and maximum coordinates across all dimensions, $v_{\min,a}$ and $v_{\max,a}$, are determined as $v_{\min,a} = \min_{j=0,\ldots,N-1;i=0,\ldots,m_j-1} v_{j,a}^i$ and $v_{\max,a} = \max_{j=0,\ldots,N-1;i=0,\ldots,m_j-1} v_{j,a}^i$, where $v_{i,a}^j \in \mathbb{R}$ is the coordinate of $\boldsymbol{v}_i^j$ along axis $a \in \{x,y,z\}$. Consider the side length $s = \max_{a \in x,y,z}(v_{\max,a} - v_{\min,a})$, the coordinates of each atom can be shifted by $(v_{\min,x}, v_{\min,y}, v_{\min,z})$ and re-scaled by $s$, bounding all $x_i, y_i, z_i \in [0,1]$.

4. **Introduction of auxiliary value and per-atom vector construction:** For each normalized atom $i$ with position $\tilde{\boldsymbol{v}}_i = (x_i, y_i, z_i)$, an auxiliary value $\sqrt{3 - x_i^2 - y_i^2 - z_i^2}$ is added, forming a 4D vector $(x_i, y_i, z_i, \sqrt{3 - x_i^2 - y_i^2 - z_i^2})$ with norm $\sqrt{3}$. The atom type $a_i$ (one-hot vector) has norm 1, yielding a per-atom vector $(x_i, y_i, z_i, a_i[1], \ldots, a_i[k], \sqrt{3 - x_i^2 - y_i^2 - z_i^2})$ with total norm $\sqrt{4} = 2$ and length $4 + k = 9$ (for $k = 5$).

5. **Concatenation, global normalization, and amplitude encoding:** Per-atom vectors are concatenated into a per-molecule vector of length $m_j \times (4 + k)$ with norm $2\sqrt{m_j}$. This vector is divided by $2\sqrt{m_j}$ to achieve unit norm. If $m_j < m_{\max} = 9$ (QM9's maximum atoms in our setting), it is zero-padded to $m_{\max} \times (4 + k) = 72$. The unit-norm vector is encoded via amplitude encoding into $|\psi\rangle = \sum_{j=0}^{2^n-1} \alpha_j |j\rangle$, where $\{\alpha_j\}$ are the vector elements (padded with zeros if needed) and $n = \lceil \log_2(m_{\max} \times (4 + k)) \rceil = 7$ qubits.

Specifically, for a molecule with $m$ atoms, the initial state is given by: $|\psi_0\rangle = \frac{1}{2\sqrt{m}} \sum_{i=0}^{m-1} \left( x_i |r_i\rangle + y_i |r_i + 1\rangle + z_i |r_i + 2\rangle + |r_i + 3 + t_i\rangle + \sqrt{3 - x_i^2 - y_i^2 - z_i^2} |r_i + k + 3\rangle \right) + \sum_{j=m(4+k)}^{2^n-1} 0 |j\rangle$, where $r_i = (k+4)i$ defines the base index for the $i$-th atom's block in the state vector, $t_i$ is the integer index corresponding to the atom type of the $i$-th atom (e.g., 0 for C, 1 for N, 2 for O, 3 for F, reflecting the one-hot encoding with a single 1 at the appropriate position), and $k = 4$ is the number of atom types.

A.7 MORE RESULTS IN LEARNING QUANTUM DISTRIBUTION

We present additional results on learning the multi-cluster quantum distribution, as described in the main text. Figure 3 illustrates the variation of the 1-Wasserstein distance between the generated test ensemble and the true ensemble with the number of training states $N$. The number of steps in the Incremental MPE is set to $K = 10$, with each unitary $V_k$ containing $L = 10$ layers (for 4-qubit states) and $L = 20$ layers (for 6-qubit states). Here, $N$ is tractable in our tasks.

We present empirical results showing that training a large number of layers simultaneously yields poor performance, whereas maintaining the same total number of layers but iteratively optimizing shallow circuits (with a small number of layers) can lead to superior results. Figure 4 illustrates the variation in the evaluation metric with the number of incremental steps in the Incremental MPE framework for learning multi-cluster quantum distributions. Here, $L = 20$ layers are trained for 2000 epochs at each incremental step, resulting in a a total of $2000 \times k$ training epochs up to the $k$-the step. We compare the result at $k = 5$ with standard training of $L = 100$ layers for $10^4$ epochs without Incremental MPE. Thus, the total number of training epochs is the same for both methods, but the Incremental MPE requires only 1/10 of the parameters compared to the standard approach. The Incremental MPE yields significantly better results, while the standard method becomes stuck during training, empirically supporting our observation that Incremental MPE provides an effective strategy for practical training.

We further investigate the behavior of the loss function in the incremental MPE framework compared to standard training. Figure 5 depicts the loss functions for learning the multi-cluster quantum

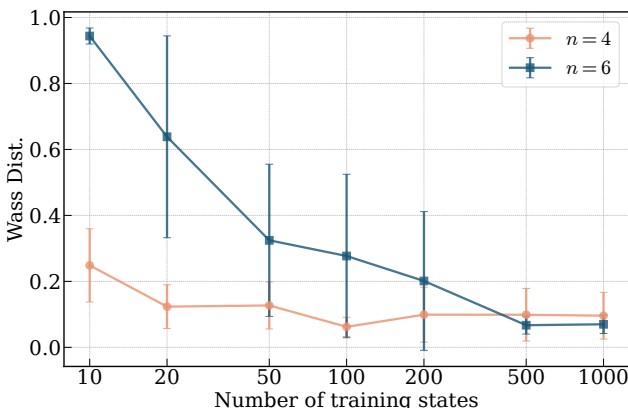

Figure 3: The 1-Wasserstein distance between the generated ensemble and the true ensemble, varying with the number of training states $K$ in the Incremental MPE framework for learning the multicluster quantum distributions of $n$-qubit quantum states ($n = 4, 6$). The solid lines represent the average accuracy over 10 trials (with error bars).

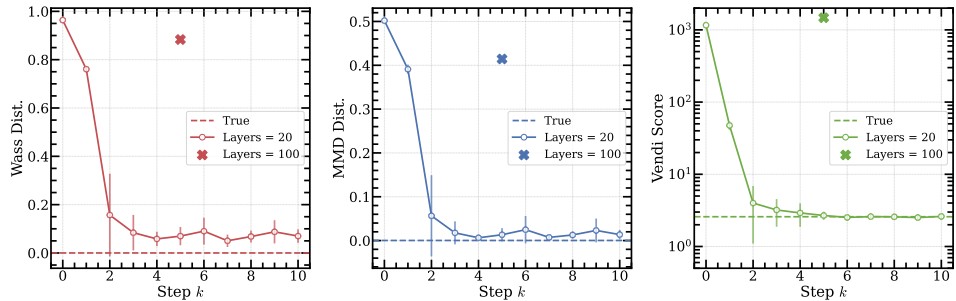

Figure 4: Evaluation metric variation with the number of incremental steps $K$ in the Incremental MPE framework for learning the multi-cluster quantum distributions. The solid lines represent the average accuracy over 10 trials (with error bars) where $L = 20$ layers are trained at each step. The cross markers represent the standard training with $L = 100$ layers without incremental steps. We plot the positions of the cross markers at step $k = 5$ to illustrate that training 20 layers over 5 steps (a total of 100 layers) is significantly better than training 100 layers at once.

distributions of $n = 6$ qubits. We compare the incremental MPE framework with $L = 10$ layers per incremental step (red curve) and direct training with $L = 100$ layers (blue curve). A barren plateaulike phenomenon is evident when training a large number of layers simultaneously (blue curve). However, even with a large total number of layers, gradually training in incremental steps enables the loss to converge to a significantly lower value. This empirically confirms the effectiveness of our incremental MPE framework in mitigating the barren plateau problem.

On learning molecular quantum states using a subset of the QM9 dataset, Fig. 6 illustrates the variation of evaluation metrics as a function of the number of incremental steps $K$ in the Incremental MPE framework. A larger $K$ generally results in lower metric values, indicating improved performance, though the metrics saturate beyond a sufficiently large $K$. Increasing the number of qubits $n_f$ in the auxiliary system $F$ can further enhance performance.

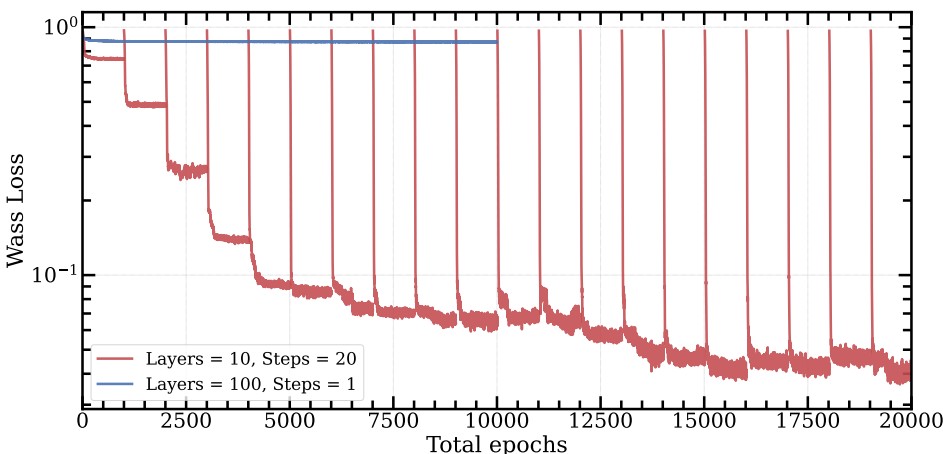

Figure 5: The training loss functions (1-Wasserstein distance) of the Incremental MPE (red) and standard training (blue) for learning the multi-cluster quantum distributions of $n = 6$ qubits. Here, we compare training Incremental MPE with $L = 10$ layers over 20 incremental steps, and standard training with $L = 100$ layers. The solid lines represent the median values at each epoch over 20 trials.

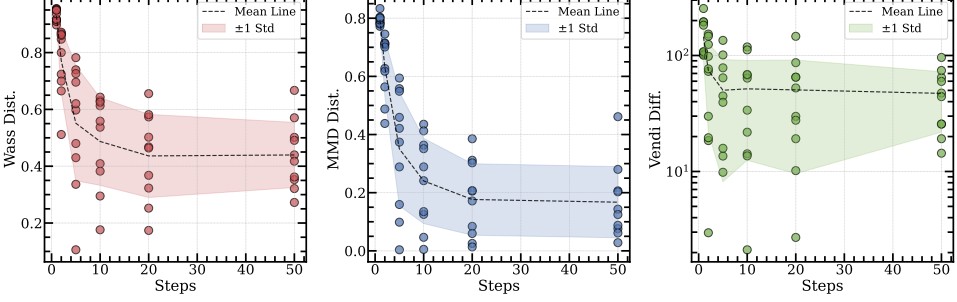

Figure 6: Evaluation metric variation with the number of incremental steps $K$ in the Incremental MPE framework for learning quantum distributions of molecular data from a QM9 subset. Circle markers represent individual trials, dotted lines indicate the mean over 10 trials, and shaded regions denote one standard deviation.

