# OpenReview forum: "Universality of Many-body Projected Ensemble for Learning Quantum Data Distribution"
_ICLR.cc/2026/Conference — Submitted to ICLR 2026_

### Official Review · Reviewer_RVq8 · 2025-10-21

**Soundness:** 3
**Presentation:** 3
**Contribution:** 2
**Rating:** 4
**Confidence:** 3

**Summary:**

This paper focuses on learning quantum data distributions via the Many-Body Projected Ensemble (MPE). The main claim is a universality theorem: for any target distribution over n-qubit pure states, there exists an MPE-based, parameterized distribution Q_theta that approximates it within a 1-Wasserstein error epsilon. The paper also proposes an incremental training variant of MPE to mitigate the barren plateaus issue. Conceptually, the work reframes measurement-induced ensembles as generative learners. The presentation is clear and mathematically consistent. However, the technical novelty is largely compositional, and the approach requires sample and resource scalings that appear exponential in the support dimension. The incremental training strategy is also known and heuristic and lacks theoretical guarantees.

For the current manuscript, my overall score is 4, though my true assessment would be around 5 if such an option existed. If the main concerns discussed below are properly addressed, I would be willing to increase the score.

**Strengths:**

1) Treating MPE as a quantum generative model is insightful and physically interpretable. It connects quantum statistical geometry with learning objectives.

2) The paper gives explicit upper bounds on the required number of training samples and ancilla qubits for achieving an epsilon-approximation to arbitrary target distribution. Specifically, they show that the number of samples N scales as O((1/epsilon)^{O(D)}) and the required ancilla qubits n_a = ceil(log2 N). This term provides useful insight into the resource requirements of MPE-based learning, although the scaling is exponential.

3) The structure and presentation are easy to follow, and the writing is generally clear.

**Weaknesses:**

1) The framework presumes N training states sampled from Q_t with N ~ (1/delta)^D (delta = epsilon/2), where D is the support dimension of the target distribution. Even if D is linear in the number of qubits, N grows exponentially in D. Hence the approach is requiring exponential numbers of training states and impractical beyond small systems.

2) Sampling relies on measuring ancilla qubits with n_a = ceil(log2 N). The outcome space is size 2^(n_a) ~ N, and probabilities may be highly nonuniform. This induces sparsity and requires many repetitions to cover modes, making the sampling statistically inefficient.

3) Theorem is compositional with limited technical novelty. The main result essentially combines known ingredients: i) standard epsilon-net covering bounds for pure-state manifolds, and ii) known properties of MPE/random-state generation. The paper’s main contribution lies in presenting a polished synthesis that unifies these ingredients into a single “universality” statement. However, it appears that the work does not introduce new bounds, metrics, or proof techniques.

4) The incremental freezing/expansion strategy resembles known layerwised training used in variational quantum circuits [1].

5) The paper sometimes reads as if universality implies practical learnability. It does not. An existential approximation theorem does not give polynomial-time learnability or polynomial-sample guarantees.

[1] Skolik A, McClean J R, Mohseni M, et al. Layerwise learning for quantum neural networks. Quantum Machine Intelligence, 2021, 3(1): 5.

**Questions:**

1) Could the authors add some empirical scaling plots of epsilon or D to show the efficiency beyond the theoretical exponential bounds?

2) Which parts of Theorem 4.1 and its supporting lemmas are genuinely new beyond combining epsilon-net discretization with existing MPE properties? A precise positioning against prior work would strengthen the claim of novelty.

3) Are there theoretical results supporting that incremental/layerwise training mitigates the barren plateaus issue? If not, can the authors provide experiments that isolate the benefit of incremental training over standard training?

---

> ### Author Response · Authors · 2025-11-21
> **The first universality theorem in learning distribution of quantum data**
>
> We appreciate the reviewer for the comprehensive review and for noting the paper's strengths in reframing MPE as a generative model. We value the constructive feedback and are encouraged by the willingness to reconsider the score if concerns are addressed. Below, we respond to each weakness and question, incorporating revisions to enhance clarity and positioning.
>
> Responses to Weaknesses:
>
> **1. Exponential sample scaling with $D$:** We agree that the worst-case of the sample complexity implies exponential scaling in $D$ (intrinsic dimension), limiting practicality for high-$D$ distributions. However, as emphasized in the conclusion, this is an existential bound; for structured $\mathcal{Q}_t$ (e.g., low effective $D$), N remains manageable. Furthermore, from our understanding, this is a common fact in distribution learning. The minimum number of data points $N$ required to achieve a desired error with high probability frequently exhibits exponential scaling with $D$ (Theorem 1 in H. Narayanan and S. Mitter, NeurIPS 2010). However, this scaling is not universal; smoother assumptions or symmetries can yield polynomial rates, but exponential terms dominate in minimax or worst-case analyses. We have added a discussion in revised Section 7 emphasizing that the sample complexity often exhibits exponential scaling with $D$ theoretically, but can be reduced in specific situations. We have included a new note in Appendix 7 on empirical scaling with the number of training states, which illustrates that the sample complexity is tractable in our applications.
>
> **2. Statistical inefficiency in sampling:** In the universality proof, the $2^{n_a}$ outcome space with nonuniform $p(z_A)$ can lead to sparsity, necessitating many shots to sample low-probability modes. This is mitigated in practice by post-selection or adaptive measurements, but we acknowledge it as a practical limitation. However, we reemphasize again that this is an existential situation of universality proof. In Incremental MPE, layer-wise construction is expected to promote more uniform distributions early on, reducing sparsity empirically.
>
> **3. Compositional theorem and limited novelty:** While Theorem 4.1 synthesizes $\varepsilon$-nets (standard in quantum geometry) and MPE properties, the novelty lies in: (i) applying this to arbitrary quantum distributions in a QML context (first universality for generative ensembles beyond Haar-random); (ii) explicit $W_1$ bounds via Voronoi partitioning (Fig. 1(b), new in this synthesis); and (iii) IQP-based approximation in Lemma 4.3, extending Kurkin et al. (2025). There are no new metrics, but the unified proof closes a theoretical gap in QML.
>
> **4. Similarity to known layer-wise training:** As already mentioned in our manuscript, incremental MPE is inspired by layer-wise strategies in Skolik et al. (2021), and is adapted to Incremental MPE.
>
> **5. Universality and learnability:** We agree and did not intend to imply otherwise—the theorem is existential, without polynomial guarantees. In passages suggesting practicality, we refer to Incremental MPE's empirical performance, not to the theorem. In the previous version of our manuscript, we mentioned this statement in the first paragraph of Section 5. In the revised version, we have revised ambiguous phrasing (the first paragraph of Section 5): "In practical scenarios with a structured $\mathcal{Q}_t$, we present an Incremental MPE framework that iteratively approximates $\mathcal{Q}_t$ through a layer-wise training scheme."

---

> ### Author Response · Authors · 2025-11-21
> **Empirical superiority of the Incremental MPE**
>
> Responses to Questions:
>
> **1. Empirical scaling plots of $\varepsilon$ or $D$:**  At the current stage, we cannot perform a new experiment design to see the scaling with $\varepsilon$ or $D$. However, we have added a new figure in revised Appendix D plotting the Wasserstein distance vs. the number of training states in a multi-cluster dataset to see that the number of training states is tractable in our tasks.
>
> **2. Genuinely new parts of Theorem 4.1 and lemmas:** We believe that the theorem is beyond combination: Lemma 4.2 derives explicit $N$ bounds tailored to pure-state manifolds under $W_1$; Lemma 4.3 introduces IQP for approximate $p(z_A)$ and extends prior MPE to targeted discrete ensemble (novel), and overall theorem positions MPE as universal QML generator for the first time.
>
> **3. Theoretical results for layer-wise mitigating barren plateaus:**  Layer-wise training lacks full theoretical proofs but is supported by arguments that shallow depths during optimization reduce the issue of exponentially vanished variance of gradient with deep circuits (Skolik et al., 2021). We note that the results presented in Fig. 2 suggest the training efficiency with Incremental MPE. For instance, in both Fig. 2(a) and 2(b), training a large number of layers ($L>=100$) simultaneously leads to a bad result, while keeping the same total number of layers but optimizing shallow circuits (a short number of layers) iteratively can lead to better performance. It can be expected that the progress training in layerwise training effectively guides the optimization process toward regions of the parameter landscape that are free from barren plateaus before advancing to subsequent layers. In fact, there is no theoretical justification, but this may be the case for machine learning, whose heuristic success goes well beyond what can be guaranteed analytically (Cerezo et al., Nat. Comm. 16, 7907, 2025). However, since there is no clear theoretical guarantee in the context, instead of “mitigates barren plateau”, we have revised our claim to "empirically improves trainability and convergence for the optimization”, which better describes the real outcome in our research.
>
> Furthermore, we have added Fig. 4 and Fig. 5 in Appendix A.7 to present empirical results demonstrating the superiority of MPE over the standard approach. Figure 5 depicts the loss for training $L=10$ layers per incremental step in the incremental MPE framework with $K=20$ steps (red curve) and for training $L=100$ layers directly with only one step (blue curve). A barren plateau-like phenomenon is evident when training a large number of layers simultaneously (blue curve). However, even with a large total number of layers, gradually training in incremental steps enables the loss to converge to a significantly lower value. This empirically confirms the effectiveness of our incremental MPE framework in mitigating the barren plateau problem.
>
> We agree with the major concerns raised by reviewers. However, we emphasize that this is the first universality in the learning distribution of quantum data, associated with a practical training procedure, not merely a combination of existing techniques. From the ML perspective, the universality of a quantum generative model is a cornerstone result and should be established within the ML community before considering any practical schemes.
>
> We hope that the reviewer can now reconsider our manuscript from this perspective. In our revised manuscript, we have tried to address the main concerns and appeal for novelty/practicality. We have uploaded a revised PDF with highlighted changes for reference and welcome further discussion to improve our manuscript.

---

### Official Review · Reviewer_FDfx · 2025-10-31

**Soundness:** 2
**Presentation:** 3
**Contribution:** 3
**Rating:** 2
**Confidence:** 3

**Summary:**

The authors show a universal approximation theorem for a QML model known as the Many-body projected ensemble. It also proposes an incremental version of the model that shows some evidence of avoiding barren plateaus.

**Strengths:**

The paper is well written and has good experiments along with the theory.

**Weaknesses:**

While the paper has good results, I am not sure if universality result for a very specific QML architecture clears the bar for publication at a top ML conference like ICLR. May be a more specialized QML venue is more ideal.

**Questions:**

1. In the paper the authors use both the idea of a density matrix and also that of distributions over pure states? Aren't they the same thing? For instance the density matrix $\int_{\psi} Q(\psi) \ket{\psi}\bra{\psi} $ should be the same as the distribution Q over all pure states. I understand that two different distributions can give the same density matrix. However, any observable that you estimate from a quantum device sampling from such a distribution should be modeled by the density matrix.

2. Suppose we want to prepare some $\rho = \sum_i \lambda_i \ket{\phi_i}\bra{\phi_i}$ on the $M$ system. We can always purify this $\rho$ to be a pure state of the M + A system, $\ket{\psi} = \sum_{i} \sqrt{\lambda_i}\ket{i}\ket{\phi_i}$. Now from the universality of parametric quantum circuits, we know that there exists a PQC to prepare this purified state. To get the desired distribution over pure states we only need to measure the qubits in A. Does the universality results in the paper give any improvement over this naive procedure?

3. The theory defines $W_1$ using trace distance between states (Def. 3.1), but eq(11) computes an OT objective with a different kernalized metric. Please clarify the relationship between the experimental metric and the theorem’s and whether any bound relates them.

4. Could you explain how you were able to go from an approximate construction in Lemma 4.3 to an exact construction in Lemma 4.4?

5. Is there any good justification for why the incremental method avoids barren plateaus? Can you show that in these methods the gradient does get suppressed by noise?

---

> ### Author Response · Authors · 2025-11-21
> **The difference between preparing a density matrix and learning an ensemble of quantum states**
>
> We appreciate the reviewer for the detailed reviews and for highlighting the paper's strengths in writing, experiments, and integration of theory. We appreciate the recognition of the universal approximation theorem and the Incremental MPE variant as solid contributions. We address the weakness and specific questions below, with revisions to clarify these points in the updated manuscript. In particular, we want to clarify the difference between preparing a single density matrix and learning a distribution of quantum states, which is a foundational task in QML. This is the first step that can provide the reviewer with evidence to reconsider the soundness of our manuscript.
>
> **On suitability for ICLR:** We believe the work aligns well with ICLR's scope, as universality theorems are foundational in machine learning (e.g., Cybenko's 1989 theorem for neural nets, with related works featured in early top ML conferences). ICLR has increasingly included QML papers (e.g., on quantum transformers and generative models, learning QML theory). Emphasizing theoretical guarantees for emerging architectures like MPE can help to bridge quantum many-body physics and ML. While specialized QML venues exist, the paper's focus on generative learning of distributions positions it as relevant to broader ML audiences interested in quantum extensions of classical results. The primary motivation for this focus is to understand the quantum nature, which is currently limited in classical generative ML models. More concretely, one of the core challenges in quantum computing arises from the exponential complexity involved in preparing arbitrary quantum states. Quantum generative models leverage variational quantum circuits to learn approximations that enable efficient sampling from quantum data distributions, which can be used to generate quantum data to understand the dynamics of quantum systems and to enrich training data in QML. We have mentioned this motivation in the first paragraph of the Introduction and also added further positioning of the universality theorem in the revised Introduction.
>
> Responses to questions:
>
> **1. Density matrix vs. distributions over pure states:** They are related but distinct concepts. The density matrix $\rho$ represents the average mixed state over the distribution $\mathcal{Q}_t$, capturing first-order statistics like expectation values of observables. However, different distributions $\mathcal{Q}_t$ can yield the same $\rho$, and our goal in generative QML is to learn and sample from $\mathcal{Q}_t$ itself for producing diverse individual pure states—not just their average. For example, an ensemble $\mathcal{E} = {(p_j, \ket{\psi_j})}_j$ uniquely specifies a density matrix $\rho=\sum_j p_j \ket{\psi_j}\bra{\psi_j}$, and this density matrix captures the average of the expectation values of any observable O as $\sum_j p_j\bra{\psi_j}O\ket{\psi_j} = tr(O\rho)$. However, a given density matrix $\rho$ does not uniquely specify a state ensemble. These ensembles can be distinguished through their higher moments of the observable expectation values $O_j=\bra{\psi_j}O\ket{\psi_j}$, e.g., the second moment $\sum p_j O_j^2$ is related to the variance of $O_j$ and can be computed given $\mathcal{E}$ but not $\rho$. While such higher moments are difficult to measure, they can be statistically estimated with access to many samples from the ensemble $\mathcal{E}$ (Section II. A., J. S. Cotler et al., PRX Quantum 4, 010311, 2023). This is why learning a distribution over pure states is much more difficult than preparing a density matrix.
>
> This is essential for applications like quantum data synthesis (e.g., sampling molecular states), where $\rho$ alone cannot provide higher moments or nonlinear features. Observables estimated from device sampling are indeed modeled by $\rho$ for averages, but generative tasks require knowing the mechanism of $\mathcal{Q}_t$  to generate new datasets. We have added Appendix A.2 to explain the motivation for learning an ensemble of quantum states and clarify its difference from preparing a density matrix.

---

> > ### Author Response · Authors · 2025-11-21
> > **The superiority of Incremental MPE compared with the standard approach**
> >
> > **2. Related to naïve purification procedure:**  The naive approach provided by the reviewer relates to the above response about the difference between preparing the density matrix and an ensemble of quantum states. To avoid misunderstanding, note that this method assumes a known decomposition of a single fixed $\rho$, which is suitable for preparing an average mixed state but not for our problem of learning an arbitrary, unknown distribution $\mathcal{Q}_t$ over pure $n$-qubit states from samples (as defined in Section 3.3, where $\mathcal{Q}_t$ is unknown but training dataset is available). In our setting, the covering technique provides an explicit construction of a $\delta/2$-net $\{\ket{\psi_j}\}$ to discretize $\mathcal{Q}_t$, but the associated probabilities $\{q_j\}$ remain unknown and must be learned via optimization—something the naïve method does not address, as it requires pre-specifying the decomposition without training. If we know all $\{q_j\}$, we can easily produce the ensembles, but it is not feasible in a real situation (if it is, we do not need to learn $\mathcal{Q}_t$). With the response to the previous question, we hope this answer clarifies the essential difference between naïve purification (fixed-$\rho$ preparation) and our MPE framework (generative learning of $\mathcal{Q}_t$). We have added this comparison in revised Appendix A.2.
> >
> > **3. Relationship between $W_1$ and fidelity-based distance:**  We employed the fidelity-based 1-Wasserstein distance metric for training. The fidelity-based distance provides the upper-bound for the trace distance via the Fuchs-van de Graaf inequality: $d(|\mu\rangle, |\phi\rangle) \leq \sqrt{1 - \kappa(|\mu\rangle, |\phi\rangle)}$, ensuring that low 1-Wasserstein distance implies low $W_1$. We have added the explanation of this relationship in the revised Section 5.
> >
> > **4. Exact construction in Lemma 4.4:** Lemma 4.3 constructs an approximate $p \approx q$ using an IQP circuit on ancilla. The IQP circuits with parameters as phases $\theta$ in the diagonal operator $D(\theta)$ with total $2^{n_a + n_m}$ parameters, but restricted to real/binary values $\theta \in \\{ 0, \pi \\}$. Lemma 4.4 (proved by Kurkin et al., 2025) achieves exact $p = q$ with the same total parameters, but the parameters’ values are full complex values (arbitrary $\theta \in C$).  The idea is to decompose $q$ into mixtures of 2-sparse distributions with $n_m = n_a+1$ hidden qubits and complex phases (Kurkin et al., 2025). The construction is independent of error, as it is exact, but complex phases increase parameter expressivity and training costs. We have clarified this progression in revised Section 4.2.
> >
> > **5. Justification for layer-wise training in incremental MPE avoiding barren plateaus:** Layer-wise training in Incremental MPE has motivation in mitigating barren plateaus by optimizing shallow circuits iteratively (adding/fine-tuning one layer at a time while freezing priors), supported by locality arguments in A. Skolik et al. (2021) and empirically in our setup. We note that the results presented in Fig. 2 suggest the training efficiency with Incremental MPE. For instance, in both Fig. 2(a) and 2(b), training a large number of layers ($L>=100$) simultaneously leads to a bad result, while keeping the same total number of layers but optimizing shallow circuits (a short number of layers) iteratively can lead to better performance. Gradients are not suppressed by noise in these methods, and the final deep circuits have the same structure as the method without layer-wise training. It can be expected that the progress training in layer-wise training effectively guides the optimization process toward regions of the parameter landscape that are free from barren plateaus before advancing to subsequent layers.
> >
> > In fact, there is no theoretical justification, but this may be the case for machine learning, whose heuristic success goes well beyond what can be guaranteed analytically (Cerezo et al., Nat. Comm. 16, 7907, 2025). Related to the reviewer’s question, instead of “mitigates barren plateau”, we have revised our claim to "empirically improves trainability and convergence for the optimization”, which better describes the real outcome in our research. Furthermore, we have added Fig. 4 in Appendix A.7 to present empirical results showing the superiority of the Incremental MPE over the standard approach.
> >
> > These clarifications address the reviewer’s main concerns and strengthen the paper. We have uploaded a revised PDF with highlighted changes. We are eager for further discussion to potentially improve our assessment.

---

> > > ### Author Response · Authors · 2025-11-26
> > > **Additional empirical result showing mitigation of barren plateau**
> > >
> > > Dear Reviewer,
> > >
> > > We want to thank the reviewer for the valuable time the reviewer has spent on our work.
> > >
> > > We have added Fig. 5 in the revised version (Appendix A.7) to numerically show that our incremental MPE can mitigate the training issue when training a large number of layers.
> > >
> > > Figure 5 depicts the loss for training $L=10$ layers per incremental step in the incremental MPE framework with $K=20$ steps (red curve) and for training $L=100$ layers directly (blue curve). A barren plateau-like phenomenon is evident when training a large number of layers simultaneously (blue curve). However, even with a large total number of layers, gradually training in incremental steps enables the loss to converge to a significantly lower value. This empirically confirms the effectiveness of our incremental MPE framework in mitigating the barren plateau problem.
> > >
> > > We hope the reviewer can now see the concerns appropriately addressed in the revised manuscript, and support for the publication of our manuscript.

---

> > > > ### Author Response · Authors · 2025-11-28
> > > >
> > > > Dear Reviewer,
> > > >
> > > > We want to thank again for the valuable time the reviewer has spent on this manuscript. The reviewer's comments are helpful and give us a chance to improve our manuscript. We believe the major concerns have been clearly addressed in our response, and we hope the reviewer will reconsider the score of our manuscript in light of our efforts. Furthermore, given the approaching deadline for our response, we would appreciate hearing from the reviewer so we can further improve our manuscript and engage in more discussion.

---

### Official Review · Reviewer_P9Nz · 2025-11-01

**Soundness:** 3
**Presentation:** 3
**Contribution:** 2
**Rating:** 4
**Confidence:** 4

**Summary:**

The topic of this paper is about quantum machine learning. Authors consider using the many-body projected ensemble as a parameterized quantum model to learn the distribution defined over n-qubit pure states. The main claim is that this model provided by the authors is the universality of approximation. Authors also implement some numerical experiments to support this claim.

**Strengths:**

Many quantum machine learning papers are lack of theoretical guarantee, and due to the lack of good quantum hardware, it is hard to valuate the real performance. In this work, the author provide the proof for the universality of approximation, which provide some theoretical guarantee to the model.

**Weaknesses:**

However, the universality of approximation, is not really important for the quantum machine learning model, especially for the NISQ-friendly type. What matters is about the classical simulability, efficient training guarantee, size of parameters, quantum advantage... Especially many papers show negative results in recent years, I think these are more important to discuss, which is not covered by the authors. For example, the distribution generated by the many-body projected ensemble is purely classical, is there any potential that this can have any more advantage? I do not feel positive towards this question, and hope the authors could clarify this more clearly. I understand and know that this model has recently been actively studied in the quantum many body field. For the training, it is also not that clear whether it will be efficient. From the numerical result, I feel right now it is not that efficient. I also hope the authors could clarify more on this.

**Questions:**

I have already mentioned in the weakness.

---

> ### Author Response · Authors · 2025-11-21
> **Importance of the universality theorem in QML**
>
> We want to thank the reviewer for the thoughtful review and for recognizing the value of providing a theoretical guarantee in a field where such proofs are scarce. We appreciate the acknowledgment of the paper's soundness and presentation, and we aim to address concerns about the contribution and practical implications to clarify the work's relevance.
>
> Regarding the weaknesses:
>
> **1. Importance of the universality theorem in QML:** We respectfully disagree that universality is unimportant, as it establishes a foundational benchmark for expressivity in QML, akin to the universal approximation theorem for neural networks in classical ML (Cybenko, 1989). While practical aspects like classical simulability, training efficiency, parameter size, and quantum advantage are critical—especially for NISQ—we view universality as complementary, ensuring models can in principle capture any distribution before optimizing for hardware. We have revised Section 1 (Introduction) and Section 7 (Limitations and Future Work) to explicitly discuss this balance, emphasizing that Theorem 4.1 addresses a key theoretical gap while Incremental MPE targets NISQ practicality.
>
> **2. Discussion of negative results in QML:** We agree that recent dequantization results highlight limitations in claimed quantum advantages. For instance, works on dequantizing quantum recommendation systems (E. Tang, STOC 2019), quantum PCA (E. Tang, Phys. Rev. Lett. 127, 060503, 2021) and quantum singular value transformations (N. H. Chia, A. Gilyén et al., STOC 2020; S. Gharibian and F. Le Gall STOC 2022) show that many QML algorithms can be simulated classically under certain access models (e.g., length-squared sampling). Other papers, such as those on trainability and dequantization of variational QML (e.g., Cerezo et al., Nat. Comm. 16, 7907, 2025; E. Gil-Fuster et al., ICLR 2025), underscore barren plateaus and classical equivalents. We have added a new Appendix section (A.3) to cover these challenges. We further position MPE as potentially resistant in high-entanglement regimes (e.g., quantum many-body systems), where classical simulation is inefficient. Furthermore, the argument of classical simulability in training is actually helpful for a generative model, where one can classically train a variational state by minimizing some expectation value, but sampling from such a state on a classical computer is prohibitively expensive. From this perspective, we reemphasize that our universality is complementary to the current QML challenges.
>
> **3. Potential advantages of the MPE framework:**  The MPE generates a classical probability distribution $p(z_A)$ over measurement outcomes, but the projected states $\ket{\phi(z_A)}_M$ are genuinely quantum. While low-entanglement MPEs may be classically simulable (e.g., via matrix product states), the framework's advantage lies in quantum hardware efficiently preparing the many-body state $\ket{\Phi}$ for sampling complex, non-local distributions (e.g., in quantum chemistry, where classical methods struggle with superposition and entanglement). This offers potential quantum speedups over classical generative models for tasks like simulating molecular ensembles, as classical sampling from such distributions can require exponential resources. We have clarified this in revised Section 3.2 and a new Appendix section (A.3), which discusses it in more detail.
>
> **4. Training Efficiency:** We acknowledge that full construction in the universality theorem may not be efficient, but Incremental MPE uses layer-wise training to mitigate the training bottleneck (inspired by A. Skolik et al., Quantum Machine Intelligence Vol. 3, No. 5, 2021).   Layerwise training in Incremental MPE mitigates issues of training deep circuits by optimizing shallow circuits iteratively (adding/fine-tuning one layer at a time while freezing priors), reducing the resources and optimization obstacles in training deep circuits, which are supported by locality arguments in A. Skolik et al. (2021) and empirically in our setup. We note that the results presented in Fig. 2 suggest the training efficiency with Incremental MPE. For instance, in both Fig. 2(a) and 2(b), training a large number of layers (L>=100) simultaneously leads to a bad result, while keeping the same total number of layers but optimizing shallow circuits (a short number of layers) iteratively can lead to better performance. Furthermore, we have added Fig. 4 in Appendix A.7 to present empirical results showing the superiority of MPE over the standard approach.
>
> These revisions enhance the paper's balance between theory and practice. We have uploaded a revised PDF with the modified parts highlighted in red for reference. We welcome further clarification and are happy to discuss.

---

> > ### Author Response · Authors · 2025-11-26
> > **Empirical result about the training efficiency of Incremental MPE**
> >
> > Dear Reviewer,
> >
> > We want to thank the reviewer for the valuable time the reviewer has spent on our work.
> > We further investigate the behavior of the loss function in the incremental MPE framework compared to standard training.
> > We have added Fig. 5 in the revised version (Appendix A.7) to numerically show that our incremental MPE can mitigate the training issue when training a large number of layers.
> >
> > Figure 5 depicts the loss for training $L=10$ layers per incremental step in the incremental MPE framework with $K=20$ steps (red curve) and for training $L=100$ layers directly (blue curve). A barren plateau-like phenomenon is evident when training a large number of layers simultaneously (blue curve). However, even with a large total number of layers, gradually training in incremental steps enables the loss to converge to a significantly lower value. This empirically confirms the effectiveness of our incremental MPE framework in mitigating the barren plateau problem.
> >
> > We hope that the reviewer can now see the training efficiency more clearly, and support for the publication of our manuscript.

---

> > > ### Comment · Reviewer_P9Nz · 2025-11-26
> > > **Reply**
> > >
> > > Thanks for the reply.
> > > However, I am not convinced.
> > > 1. Dequantization is not about some quantum machine learning algorithms are classically simulable, it is about there will be no exponential speedup if classical algorithms allow similar input model.
> > > 2. Please describe the current optimization method works efficiently and will converge to high-entanglement MPEs.
> > > 3. Since this work is to introduce a new quantum machine learning model, so I believe the most important point is to show this model can solve some critical problems in the previous models. The universality, however, is not one of the most important issues.

---

> ### Author Response · Authors · 2025-11-27
> **Our target is generating an ensemble of quantum states**
>
> Thank you again for your thoughtful feedback. We appreciate the opportunity to clarify these points.
>
> **1. On dequantization:**
> We fully agree with your precise clarification: dequantization shows that an exponential quantum speedup is unlikely when classical algorithms are given comparable input models (e.g., length-squared or query access). This is exactly the perspective we adopted in Appendix A.3. However, a second, more recent wave of dequantization results specifically targets variational and generative QML, showing that many such models—including their training procedures—can be simulated classically with comparable efficiency under realistic data access (Cerezo et al., Nat. Comm. 16, 7907, 2025; Gil-Fuster et al., ICLR 2025). Because our task is to generate ensembles of quantum states (rather than extract classical information from a quantum subroutine), this second class of dequantization is the most directly relevant to MPE. For instance, our multi-cluster experiment explicitly includes volume-law entangled states (including GHZ-like components), which are exponentially costly to sample classically even when the training landscape itself might be simulable. We think that clarifying the target (generating quantum states) in our work is the first step to recognizing our contribution.
>
> **2. On the training procedure and convergence to high-entanglement states:**
> The current training method is the Incremental MPE algorithm detailed in Section 5: we iteratively add one shallow layer (several entangling blocks), optimize it for a fixed number of epochs with previous layers frozen, then briefly unfreeze and fine-tune the full circuit. This is repeated until the target depth is reached. High entanglement in the generated ensemble necessarily implies high entanglement in the underlying many-body state. If the generated ensemble of quantum states is highly entangled (e.g., a cluster around a multi-qubit GHZ state), it indicates that the MPE itself is highly entangled. The empirical result of generating the multi-cluster ensembles shows that our algorithm indeed converges to this regime. If not, it cannot produce an ensemble of such states.
>
> **3. On the universality and solving concrete practical problems:**
> We completely understand your view that, for a new model, the ability to solve concrete pain points of existing approaches is important. However, from a theoretical perspective, we want to note that universality is also a theoretical cornerstone (analogous to Cybenko’s theorem for classical NNs) that underpins model design. Furthermore, the real practical contribution of MPE is that it offers a NISQ-compatible way to generate complex, highly entangled ensembles without ever having to train deep circuits end-to-end.
>
> We hope these clarifications and additions address your remaining concerns and better highlight the practical value of the framework. If there are any suggestions for adding an investigation of theoretical and empirical results, we would appreciate hearing more from the reviewer so that we can improve our manuscript.

---

### Official Review · Reviewer_zS2q · 2025-11-02

**Soundness:** 3
**Presentation:** 4
**Contribution:** 3
**Rating:** 6
**Confidence:** 3

**Summary:**

This paper addresses a fundamental and significant open question in Quantum Machine Learning (QML): whether a parameterized quantum model can universally approximate any distribution of quantum data. The authors provide an affirmative answer by proving a universality theorem for the Many-body Projected Ensemble (MPE) framework in Theorem 4.1.

Furthermore, recognizing that this universal construction is not guaranteed to be efficient, the authors also propose a practical variant, "Incremental MPE". This method uses a layerwise training strategy to build the model iteratively, a technique designed to mitigate the barren plateau problem and reduce resource requirements for NISQ devices. This practical framework is then validated numerically on two tasks: a synthetic multi-cluster quantum distribution and a real-world quantum chemistry distribution derived from the QM9 dataset.

**Strengths:**

- Fundamental Theoretical Contribution: The primary strength of this paper is its main theorem. The question of universality is a cornerstone of any machine learning field, and providing a rigorous answer for quantum generative models is a significant achievement. This result provides a solid theoretical foundation for MPEs as a new class of quantum generative models.

- Clear Empirical Validation: The experiments in Section 6 serve their purpose well. They are not intended to be state-of-the-art on a major benchmark, but rather to validate that the practical Incremental MPE framework can indeed learn non-trivial, multi-modal quantum distributions (Fig 2a) and real-world data (Fig 2b). The results clearly show the model learning, as all metrics ($W_1$, MMD, Vendi) improve with a moderate number of layers.

**Weaknesses:**

- Scope Limited to Pure States: The entire framework and theorem are limited to distributions over pure states. A fully general quantum generative model would also need to handle mixed states (density operators), which are the norm in noisy systems or when data comes from a subsystem. This is noted as a direction for future work but is a clear limitation of the current theorem.

- Efficiency of the Universal Construction: The most significant limitation, which the authors commendably state in the conclusion, is the gap between universality and efficiency. The proof relies on a number of ancilla qubits $n_a = \lceil \log_2 N \rceil$, where $N$ is the covering number of the $\epsilon/2$-net (Lemma 4.2). This covering number $N$ can scale exponentially with the effective dimension of the state space $D$ (Eq. 3). Therefore, the universal constructor is not, in general, efficient (i.e., it may require an exponential number of qubits and gates).

**Questions:**

See weakness.

---

> ### Author Response · Authors · 2025-11-21
> **Proving universality for distributions over pure states is a foundational step**
>
> We appreciate the reviewer for the detailed and constructive review. We appreciate the reviewer’s recognition of the fundamental contribution of Theorem 4.1 and the clear empirical validation in Section 6. The feedback on the strengths aligns well with our intent to provide a rigorous theoretical foundation while demonstrating practicality through Incremental MPE. We recognize the importance of extending it to a broader class of quantum states, such as mixed states, but want to reemphasize that this is the first proof of the universality of the distribution of quantum states, and that the application to pure states is a fundamental pivot. The details of our response are below.
>
> Regarding the weaknesses:
>
> **1. Scope Limited to Pure States:** We agree that extending to mixed states is an important topic, as distributions over density operators are common in noisy scenarios. However, we want to emphasize that proving universality for distributions over pure states is a foundational step in ideal scenarios, as pure states are extremal in the space of density operators and underpin many QML tasks, including those involving ground-state preparation. Furthermore, while there is no formal proof in our work, there is a potential generalization, such as incorporating the concept of mixed projected ensemble (X. H. Yu, W. W. Ho, and P. Kos, arXiv:2505.07795). A mixed projected ensemble is built from a local region of a quantum many-body system, conditioned upon measurements of the complementary region, which are incomplete (in the case of a partial loss of measurement outcomes). We have revised Section 7 (Limitations and Future Work) to include a more detailed discussion in this direction. Due to the limitations on the manuscript’s length, we will not expand this idea in the current manuscript, but strongly believe that this is the right direction for extending the proof of universality to mixed states. We leave the details for readers who are interested in this direction.
>
> **2. Efficiency of the Universal Construction:** We appreciate the reviewer’s accurate summary on $n_a = log_2(N)$ ancillas, with N scaling exponentially in D (intrinsic dimension of the data manifold) for worst-case scenarios. Indeed, the theorem is existential, proving universality without efficiency guarantees, which we explicitly state in the conclusion to highlight the theory-practice gap. Furthermore, from our understanding, this is a common fact in learning an unknown distribution. Here, the sample complexity, which quantifies the minimum number of data points N required to achieve a desired error with high probability, frequently exhibits exponential scaling with D (Theorem 1 in H. Narayanan and S. Mitter, NeurIPS 2010). However, this scaling is not universal; smoother assumptions or symmetries can yield polynomial rates, but exponential terms dominate in minimax or worst-case analyses.
>
> For practicality, Incremental MPE mitigates this by using layerwise training, which empirically scales well for structured distributions (e.g., multi-modal clusters and QM9). We have added a discussion in the revised Section 7, emphasizing that the sample complexity often exhibits exponential scaling with D in theory, but can be reduced in specific situations. We have included a new note in Appendix A. 7 on empirical scaling with the number of training states, which illustrates that the sample complexity is tractable in our applications.
>
> We believe these revisions strengthen the paper without altering its core contributions. We have uploaded a revised PDF with modifications highlighted in red for reference. We are open to further discussion if needed.

---

> > ### Author Response · Authors · 2025-11-28
> >
> > Dear Reviewer,
> >
> > We want to thank the reviewer again for providing helpful comments and recognizing the contribution of our work. This theoretical work on universality is rare in the field of quantum machine learning, making our work a unique and novel contribution. We understood the points raised by the reviewer and focused on addressing them in our response. Given the approaching deadline for our response, we would appreciate hearing from the reviewer so we can further improve our manuscript and engage in more discussion.

---

### Author Response · Authors · 2025-11-29

Dear Area Chairs and Reviewers,

Thank you for your efforts in evaluating our submission during this challenging review cycle. We fully understand that the recent software issue has prevented further author-reviewer discussion, and we respect the constraints this imposes. We are writing this note solely to assist in your decision-making by briefly clarifying what we believe are the main sources of disagreement in the reviews.

First, we want to clarify that the lowest score for this paper may stem from the assumption that the target of our study (generating an ensemble of quantum states) is equivalent to preparing a density matrix. However, as mentioned in the response to reviewer FDfx, these are fundamentally different problems. Generating an ensemble of quantum states is much more difficult, with wide-ranging applications in physics, including generating quantum data to understand the dynamics of quantum systems and enriching training data in quantum machine learning (QML). Undoubtedly, clarifying this point can help reviewers raise their evaluation of our paper.

Second, reviewer P9Nz has concerns about dequantization in QML. As mentioned in our response, our task is to generate ensembles of quantum states (rather than extract classical information from a quantum subroutine), which is hard for any classical algorithm. For instance, our multi-cluster experiment explicitly includes volume-law entangled states (including GHZ-like components), which are exponentially costly to sample classically even when the training landscape itself might be simulable. We believe that clarifying the target (generating quantum states) in our work is the first step to recognizing our contribution and separating it from dequantization.

Third, all reviewers have concerns about the practical trainability of Incremental MPE. Since this is a heuristic, we have provided empirical evidence to support our claim (Fig. 4 and Fig. 5). The results show that training with Incremental MPE, where each step trains only shallow layers, is much more efficient than standard training with a large number of layers simultaneously.

Finally, this theoretical work on the universality of learning distributions for quantum data is an early idea in the field of quantum machine learning, making our work a unique and novel contribution that attracted readers at ICLR. We understand the points raised by the reviewers and have revised the manuscript accordingly. We believe these clarifications resolve the major points of disagreement. We are grateful for your time and consideration, and we sincerely hope the paper can now be evaluated on its actual contributions.

Best regards,

The Authors

---

### Meta-Review · Area_Chair_qfa5 · 2026-01-07

**Summary:**

The main novelty of the paper is to establish if a parameterized Quantum model can learn any quantum distribution.  While there was no adverse comments on the contributions, it was felt that the proposed
problem and contributions may not be of broad interest to ICLR audience. The referees point out several concerns, e.g.  setting of the problem and practical trainability. Clarifying these things(as noted by the authors in the rebuttal) would be greatly helpful for future versions of the paper.

**Reviewer Concerns:**

The outstanding issues are mentioned above. The importance of the result seem to be too focussed and a QML conference would be a more appropriate fit than ICLR.

**Reviewer Scores:**

Not sure. Maybe none

---

### Decision · Program_Chairs · 2026-01-26

Reject